# Offline Policy Interval Estimation without Sufficient Exploration or Realizability

## Abstract

We study the problem of offline policy evaluation (OPE), where the goal is to estimate the value of given decision-making policy without interacting with the actual environment. In particular, we consider the interval-based OPE, where the output is an interval rather than a point, indicating the uncertainty of the evaluation. The interval-based estimation is especially important in OPE since, when the data coverage is insufficient relative to the complexity of the environmental model, any OPE method can be biased even with infinite sample size. In this paper, we characterize such irreducible biases in terms of the discrepancy between the target policy and the data-sampling distribution, and show that the marginal-importance-sampling (MIS) estimator achieves the minimax bias with an appropriate importance-weight function. Motivated with this result, we then propose a new interval-based MIS estimator that asymptotically achieves the minimax bias.

## 1 Introduction

The offline policy evaluation (OPE) is the art of estimating the value of given decision-making policies based on offline datasets without interacting with the actual environment. Since the interaction with the environment is often infeasible or expensive in many real-world applications, it is better to evaluate the value offline rather than online.

In the literature, it is understood from theoretical perspectives that there are two fundamental conditions for OPE to be successful: *sufficient exploration*, the coverage of the data-sampling distribution over the state-action space relative to the target policy, and *realizability*, the knowledge of correct environmental model with bounded complexity. In particular, if neither of these two conditions are met in a certain manner, it is known that OPE is never sample efficient, i.e., it takes prohibitively large sample to make the estimation reasonably accurate (Wang et al., 2020; Zanette, 2021). In practice, given a problem instance of OPE, consisting of an environment and a dataset, it is difficult to confirm that these conditions hold or to modify the problem instance so that these conditions hold, making the existing theoretical guarantees less practical. Towards practical OPE, we set our research objective to develop a theoretically-sound value estimator without assuming these two conditions.

Towards our objective, we first analyze the statistical performance of OPE methods when the two assumptions do not hold (Section 4). The key quantity is the information-theoretic worst-case bias of the value estimator (Eq. (5)) and its minimum termed the *minimax bias* (Eq. (6)), which is positive when there exist multiple indistinguishable environments, given only a problem instance of OPE. In fact, we show that the minimax bias can be non-zero if we do not assume the two conditions (Corollary 4.2). It suggests that, without the two assumptions, there exists a problem instance that any point-based value estimator is not reliable.

Given the existence of irreducible bias, we propose an alternative formulation of offline policy evaluation called *minimax-bias offline policy interval estimation (minimax-bias OPI)*, where the objective is to estimate the shortest possible interval containing the true value, instead of a point estimate (Section 5). Since our characterization of the minimax bias allows us to define the optimal interval (Definition 5.1), the minimax-bias OPI is formulated as a problem to estimate the optimal interval (Problem 5.1).

We provide a theoretical foundation to solve the minimax-bias OPI based on the marginal importance sampling estimator (Section 6). The key result is that the optimal importance weight mini-

mizing the distributional Bellman residual (DBR) allows us to construct an approximately optimal interval (Theorem 6.3). This illustrates that our problem setting is well-posed and can be solved under realistic assumptions if we can solve the minimization of DBR. Accordingly, we develop a novel algorithm in Section 7 to find the best importance weight function, which results in an interval estimator applicable even if the two fundamental conditions do not hold (Theorem 7.7).

Before proceeding to these results, we introduce basic mathematical notation in the rest of this section, review the related work in Section 2, and introduce the useful OPE-specific notation in Section 3.

**Mathematical notation.** Let $I$ denote the identity operator and let $a \vee b := \max\{a, b\}$ and $a \wedge b := \min\{a, b\}$ denote the maximum and minimum operators for $a, b \in \mathbb{R}$, respectively.

Let $\mathcal{X}$ be a metric space with Borel algebra $\Sigma$. Let $\mathscr{B}(\mathcal{X})$ and $\mathscr{C}(\mathcal{X})$ be the spaces of the real-valued measurable bounded functions and the continuous functions on $\mathcal{X}$, respectively, both of which is equipped with the uniform norm $\|f\|_\infty := \sup_{x \in \mathcal{X}} |f(x)|$. Let $\mathscr{M}(\mathcal{X})$ denote the space of the finite signed measures on the same space $\mathcal{X}$, equipped with the total variation (TV) norm $\|P\|_{\mathrm{TV}} := \sup_{E_+ + E_- = \mathcal{X}} \{P(E_+) - P(E_-)\}$. In particular, let $\delta_x \in \mathscr{M}(\mathcal{X})$, $x \in \mathcal{X}$, denote Dirac's delta measure. For any $f \in \mathscr{B}(\mathcal{X})$ and any $P \in \mathscr{M}(\mathcal{X})$, let $\langle f, P \rangle := \int f(x) dP(x)$ be a shorthand for the (signed) expectation of $f$ with respect to $P$. Let $\odot$ denote the importance-weighting operation given by $d(f \odot P)(x) := f(x) dP(x)$, $f \in \mathscr{B}(\mathcal{X})$, $P \in \mathscr{M}(\mathcal{X})$. Let $L^1(P)$ be the space of the functions integrable with respect to $P \in \mathscr{M}(\mathcal{X})$, i.e., $\|f \odot P\|_{\mathrm{TV}} < \infty$.

Let $\mathscr{L}(\mathcal{V})$ denote the set of the bounded linear operator on a normed vector space $\mathcal{V}$. For any $A \in \mathscr{L}(\mathscr{M}(\mathcal{X}))$, let $A^* \in \mathscr{L}(\mathscr{B}(\mathcal{X}))$ denote the conjugate operator such that $\langle A^* f, P \rangle = \langle f, AP \rangle$ for $f \in \mathscr{B}(\mathcal{X})$ and $P \in \mathscr{M}(\mathcal{X})$.

## 2 RELATED WORK

The problem of estimating the interval containing the true value has been known as *offline policy interval estimation* (OPI). This section reviews the existing studies on OPI by dividing the previous OPI methods into two categories: non-asymptotic and asymptotic methods (see Table 1 for the summary of comparison). We also discuss our contribution to the literature.

The non-asymptotic methods typically put their emphasis on the validity of the interval with any finite sample size, where intervals are valid if they contain the true value $J(\pi)$. For instance, Feng et al. (2020; 2021) compute intervals that contain the true policy value with high probability, under the realizability of the policy Q-functions $q_\pi$. Jiang and Huang (2020) also proposed an interval estimator with validity under more relaxed realizability condition that either the policy Q-function $q_\pi$ or the marginal density ratio function $w_\pi$ is realizable. One limitation of this approach is the theoretical understanding on the tightness of the interval is often unclear or partial. Another limitation of this approach is that they tend to require the realizability with *known complexity*. This requirement is not desirable for practical use; if we used a too complex hypothesis class such as a reproducing kernel Hilbert space with infinite radius, the resultant interval would be trivial, and thus, non-informative.

The asymptotic methods focus on the asymptotically dominant term of the uncertainty in the large sample limit, which typically allows us to theoretically understand their behavior, especially the tightness, in depth. For instance, Kallus and Uehara (2020); Shi et al. (2021) gave confidence interval estimators that achieve the efficiency lower bound. The bootstrap estimators (Hao et al., 2021) also enable us to compute the asymptotically exact confidence intervals in a more flexible manner. One major limitation is that they assume both the sufficient exploration and the realizability conditions of $q_\pi$ and $w_\pi$ hold, which can be hardly validated in real-world applications. These assumptions are essential to their analyses because they focus on estimation of the asymptotic *variance* of order $O(n^{-1/2})$, assuming that the *bias* is negligible. Therefore, these methods are not applicable to our setting where the asymptotic *bias* of order $O(1)$ dominates the asymptotic *variance*.

In this study, we take the asymptotic approach, but with a focus on the estimation of the *bias* rather than the *variance*, because the bias is dominant in our setting where the sufficient exploration and the realizability do not hold at all. Our contributions are threefold. First, we characterize the theoretical lower bound of the asymptotic bias through the asymptotic analysis, which serves as a theoretical

Table 1: Comparison of OPI methods. $q_\pi$ and $w_\pi$ denote the Q-function and the marginal density ratio function, respectively, $w_\pi^\sharp$ denotes a generalization of $w_\pi$ for the insufficient exploration setting.

| Method | Assumptions | | | | Guarantee |
|---|---|---|---|---|---|
| | **Asymptotic** | **Exploration** | **Realizability** | **Complexity** | |
| BONDIC (Feng et al., 2021) | — | — | $q_\pi$ | Known | Valid |
| MVI (Jiang and Huang, 2020) | — | — | $q_\pi$ or $w_\pi$ | Known | Valid |
| DRL (Kallus and Uehara, 2020) D2OPE (Shi et al., 2021) | Yes | Yes | $q_\pi$ and $w_\pi$ | — | Efficient |
| Ours | Yes | — | $\overline{\phantom{w}}$ $w_\pi^\sharp$ | — | Valid Optimal |

foundation of OPE without sufficient exploration or realizability assumptions. Second, without the two assumptions, we develop an interval estimation method that outputs an asymptotically valid interval, that is, an interval that contains the true value in the large sample limit. Third, under the realizability condition of the generalized marginal density ratio function $w_\pi^\sharp$, we show that the estimated interval is optimal.

## 3 PRELIMINARIES

We first introduce our formulation of reinforcement learning and offline policy evaluation. Then, we introduce two fundamental concepts in RL, a Q-function and an occupancy measure, along with shorthand notation for them.

**Offline policy evaluation.** Let $\mathcal{X} := \mathcal{S} \times \mathcal{A}$ be a compact Hausdorff space representing the state-action space of the system with $|\mathcal{X}| < \infty$.[1] Let $\mathcal{M} := (\iota, T, R)$ be the Markov decision process (MDP) of environment on $\mathcal{X}$, where $\iota \in \mathscr{M}(\mathcal{S})$ is the initial state distribution, $T : \mathcal{X} \to \mathscr{M}(\mathcal{S})$ is the transition dynamics and $R : \mathcal{X} \to \mathscr{M}([-1, 1])$ is the conditional reward distribution. Let $\pi : \mathcal{S} \to \mathscr{M}(\mathcal{A})$ be the target policy. Then, the *value* $J(\pi)$ of $\pi$ with respect to $\mathcal{M}$ is given by the $\gamma$-discounted expected average reward

$$J(\pi) \equiv J_\mathcal{M}(\pi) := \mathbb{E}^{\mathcal{M},\pi} \left[ (1 - \gamma) \sum_{t=1}^{\infty} \gamma^{t-1} r_t \right],$$

where $\gamma \in (0, 1)$ is a discounting factor and $\mathbb{E}^{\mathcal{M},\pi}$ denotes the expectation with respect to the Markov chain generated with $a_t \sim \pi(s_t), r_t \sim R(s_t, a_t), s_{t+1} \sim T(s_t, a_t)$ for all $t \geq 1$ and $s_1 \sim \iota$.

In offline policy evaluation, we are given a dataset $\mathcal{D} := (\mathcal{D}_\iota, \mathcal{D}_{T,R})$ as input, where $\mathcal{D}_\iota := \{s_{\iota,j}\}_{j=1}^{n}$ is a set of initial states and $\mathcal{D}_{T,R} := \{(x_i, s_i', r_i)\}_{i=1}^{n}$ is a set of transition records sampled from

$$dG_{\mathcal{M},\beta}(\mathcal{D}) := \prod_{j=1}^{n} d\iota(s_{\iota,j}) \cdot \prod_{i=1}^{n} d\beta(x_i) dT(s_i'|x_i) dR(r_i|x_i),$$

where $\beta \in \mathscr{M}(\mathcal{X})$ is an arbitrary state-action-sampling distribution.[2] Then, an instance of the offline policy evaluation (OPE) is identified by the quadruple $\mathcal{P} := (\mathcal{M}, \beta, \pi, \gamma)$ and formalized as follows.

**Problem 3.1** (Offline policy evaluation, OPE). *Given* $(\mathcal{D}, \pi, \gamma)$ *where* $\mathcal{D} \sim G_{\mathcal{M},\beta}$, *estimate* $J(\pi)$.

**Q-function and occupancy measure.** Let $\iota_\pi \in \mathscr{M}(\mathcal{X})$ and $T_\pi \in \mathscr{L}(\mathscr{M}(\mathcal{X}))$ be the initial state-action distribution and the state-action transition operator associated with $\pi$ such that $d\iota_\pi(s, a) := d\iota(s)d\pi(a|s)$ and $d(T_\pi P)(s, a) := \int dT(s|x)d\pi(a|s)dP(x)$ for $s \in \mathcal{S}$, $a \in \mathcal{A}$ and $P \in \mathscr{M}(\mathcal{X})$,

---

[1]This includes the cases where $\mathcal{S}$ and $\mathcal{A}$ are finite or compact subsets of finite-dimensional Euclidean spaces.

[2]For simplicity, we assume the sample sizes of $\mathcal{D}_\iota$ and $\mathcal{D}_{T,R}$ are the same. The generalization with different sample sizes is possible with minor modification.

respectively. Also let $\rho \in \mathscr{B}(\mathcal{X})$ be the expected reward function such that $\rho(x) := \int r \mathrm{d}R(r|x)$ for $x \in \mathcal{X}$. Then, the value $J(\pi)$ is rewritten as

$$J(\pi) = \langle \rho, \Gamma_\pi \iota_\pi \rangle = \langle \Gamma_\pi^* \rho, \iota_\pi \rangle = \langle \rho, \mu_\pi \rangle = \langle q_\pi, \iota_\pi \rangle, \tag{1}$$

where $\Gamma_\pi := (1-\gamma) \sum_{t=1}^{\infty} (\gamma T_\pi)^{t-1} \in \mathscr{L}(\mathscr{M}(\mathcal{X}))$ is the accumulation operator, $\mu_\pi := \Gamma_\pi \iota_\pi \in \mathscr{M}(\mathcal{X})$ is the normalized occupancy measure of $\pi$ (henceforth *the occupancy measure*), and $q_\pi := \Gamma_\pi^* \rho \in \mathscr{B}(\mathcal{X})$ is the normalized Q-function of $\pi$ (henceforth *the Q-function*). Note that we have $\|q\|_\infty \leq 1$ and $\|\mu_\pi\|_{\mathrm{TV}} = 1$ thanks to the normalization.

**Two Bellman equations.** One of the essential difficulties of OPE lies in the fact the direct estimation of the accumulation operator $\Gamma_\pi$ (and hence $\mu_\pi$ and $q_\pi$) is intractable due to the infinite sum. The Bellman equation is useful to mitigate this problem. Here, we introduce two variants of the Bellman equation, the *functional and distributional Bellman equations*, given by

$$\rho = \Delta_\pi^* q_\pi, \qquad\qquad \iota_\pi = \Delta_\pi \mu_\pi, \tag{2}$$

where $\Delta_\pi := \Gamma_\pi^{-1} = (I - \gamma T_\pi)/(1-\gamma)$ is the difference operator. Note that, in the Bellman equations, both $q_\pi$ and $\mu_\pi$ are *uniquely* characterized via more directly estimatable quantities $(\rho, T_\pi)$ and $(\iota_\pi, T_\pi)$, respectively.

The errors of the Bellman equations are referred to as the Bellman residuals. In particular, the distributional Bellman residual (DBR) is given by

$$\mathcal{R}_\pi(w) := \iota_\pi - \Delta_\pi(w \odot \beta) \in \mathscr{M}(\mathcal{X}),$$

which plays an important role in our analysis.

**Empirical estimates.** Finally, we introduce the empirical estimates of $(\iota_\pi, T_\pi, \rho, \beta)$ based on the dataset $\mathcal{D}$ as follows. For all $P \in \mathscr{M}(\mathcal{X})$ and $x \in \mathcal{X}$,

$$\hat{\iota}_\pi := \frac{1}{n} \sum_{j=1}^{n} \delta_{x_{\iota,j}}, \quad \hat{T}_\pi P := \sum_{i=1}^{n} \frac{\delta_{x_i'}}{N(x_i)} P(\{x_i\}), \quad \hat{\rho}(x) := \frac{1}{N(x)} \sum_{i:x_i=x} r_i, \quad \hat{\beta} := \frac{1}{n} \sum_{i=1}^{n} \delta_{x_i}, \tag{3}$$

where $N(x) := 1 \vee |\{i : x_i = x\}|$ is the data-counting function (with the zero-division safeguard) and $x_{\iota,i} := (s_{\iota,i}, a_{\iota,i})$ and $x_i' := (s_i', a_i')$ are the state-action pairs associated with additional samples $a_{\iota,i} \sim \pi(s_{\iota,i})$ and $a_i' \sim \pi(s_i')$, respectively.

Throughout this paper, we employ the conventional marginal importance sampling (MIS) estimator (Liu et al., 2018; Xie et al., 2019) to estimate the value in offline. The MIS estimator associated with a weight function $w \in \mathscr{B}(\mathcal{X})$ is given by

$$\hat{J}(w) := \langle \hat{\rho}, w \odot \hat{\beta} \rangle. \tag{4}$$

The MIS estimator is justified if the weight function $w$ is equal to the marginal density $w_\pi := \frac{\mathrm{d}\mu_\pi}{\mathrm{d}\beta}$ (assuming it exists) since, in that case, the MIS estimator is unbiased, $\mathbb{E}[\hat{J}(w_\pi)] = J(\pi)$, according to (1).

Note, however, that $w_\pi$ does not exist when the exploration is insufficient, $\beta \not\gg \mu_\pi$, and the unbiasedness cannot be guaranteed in general. Two natural questions thus arise: Does the MIS estimator still enjoy any theoretical guarantee in such a general setting? If so, what is the best weight function $w$? In short, the answer to the first question is affirmative and the answer to the second question is one of the main contributions of this work.

## 4 IRREDUCIBLE BIAS IN OFFLINE POLICY EVALUATION

In this section, we theoretically analyze the statistical performance of OPE methods without sufficient exploration or realizability assumptions. As a result, we will show that any OPE method must incur an irreducible bias that never disapears even when the sample size goes infinity. Given such a negative result, we instead propose a novel problem setting called the *minimax-bias OPI*, where the

goal is to estimate the interval that contains the true value and is as short as possible. The proposed problem setting is expected to be solved without sufficient exploration or realizability assumptions, and thus, will be of practical use.

To study the statistical performance of OPE methods, we introduce the notion of the *minimax bias* of the point-based estimators. Let $\hat{J}$ be *any* random variable representing a point-based OPE estimator. Then, the information-theoretic worst-case bias of $\hat{J}$ is given by

$$\epsilon[\hat{J}] \equiv \epsilon[\hat{J}; \mathcal{P}] := \sup_{(\mathcal{M}', \beta) \sim (\mathcal{M}, \beta)} \left| J_{\mathcal{M}'}(\pi) - \mathbb{E}\hat{J} \right|, \tag{5}$$

where the equivalence $\sim$ is defined by the equality with respect to the corresponding distributions of the dataset, *i.e.*, $G_{\mathcal{M}', \beta} = G_{\mathcal{M}, \beta}$. If there exist equivalent environments $\mathcal{M}$ and $\mathcal{M}'$ that result in the different policy values $J_{\mathcal{M}}(\pi) \neq J_{\mathcal{M}'}(\pi)$ yet indistinguishable from the dataset, the worst-case bias $\epsilon[\hat{J}]$ is inevitable without an additional source of information, i.e., domain knowledge. The minimax bias is then defined as the minimum possible worst-case bias of OPE,

$$\epsilon_\star(\pi) \equiv \epsilon_\star(\pi; \mathcal{P}) := \inf_{\hat{J}} \epsilon[\hat{J}; \mathcal{P}], \tag{6}$$

which can be thought of as a characteristic of the problem $\mathcal{P}$ indicating its hardness in terms of the irreducible uncertainty even with infinitely large sample. In fact, there exists the unique $\hat{J}$ achieving the infimum and we refer to it as the optimal point estimator $J_\star(\pi)$. Our main objective is to understand the minimax bias in various settings.

To this end, we introduce a novel concept, the projection of the occupancy measure $\mu_\pi$ with respect to $\beta$. Let $\Pi_\beta$ be the projection operator onto the support of $\beta$ such that $\Pi_\beta P = \chi_\beta \odot P$, $P \in \mathcal{M}(\mathcal{X})$ and $\chi_\beta(x) = \mathbf{1}\{x \in \text{supp}\,\beta\}$.

**Definition 4.1** (Projected occupancy measure and its importance weight). *We refer to*

$$\mu_\pi^\sharp := (1 - \gamma)\Pi_\beta \sum_{t=0}^{\infty} (\gamma T_\pi \Pi_\beta)^t \iota_\pi \tag{7}$$

*as the projected occupancy measure of $\pi$. Correspondingly, we also refer to*

$$w_\pi^\sharp := \frac{\mathrm{d}\mu_\pi^\sharp}{\mathrm{d}\beta}$$

*as the projected importance weight of $\pi$ with respect to $\beta$.*

Note that $w_\pi^\sharp$ can be thought of as an extension of $w_\pi$ in the sense it is always well-defined and $w_\pi^\sharp = w_\pi$ whenever $w_\pi$ exists. On the other hand, $\mu_\pi^\sharp$ can be thought of as the *known component* of $\mu_\pi$ since it is always identifiable given $G_{\mathcal{M}, \beta}$, thanks to the projection $\Pi_\beta$, and $\mu_\pi^\sharp = \mu_\pi$ whenever $\mu_\pi$ is also identifiable.

We now present our main result, which discovers close relationships between the minimax bias $\epsilon_\star(\pi)$, the MIS estimator $\hat{J}(w)$, the DBR $\mathcal{R}_\pi(w)$ and the projected importance weight $w_\pi^\sharp$.

**Theorem 4.1.** *For all $w \in \mathcal{B}(\mathcal{X})$, we have*

$$\epsilon_\star(\pi) \leq \epsilon\left[\hat{J}(w)\right] \leq \|\mathcal{R}_\pi(w)\|_{\text{TV}} \leq \epsilon_\star(\pi) + \frac{1 + \gamma}{1 - \gamma}\|w - w_\pi^\sharp\|_{L^1(\beta)}. \tag{8}$$

*Proof (sketch).* The most nontrivial part is the last inequality, which follows from the constructive proof of

$$\epsilon_\star(\pi) \geq \|\mathcal{R}_\pi(w_\pi^\sharp)\|_{\text{TV}}. \tag{9}$$

In particular, we construct two worst-case environments $\mathcal{M}_\pm$ under the constraint $(\mathcal{M}_\pm, \beta) \sim (\mathcal{M}, \beta)$. Roughly speaking, the environments are constructed to have a special state $\perp$ in the under-explored region of $\mathcal{X}$ absorbing all the transition to that region, and have the extreme reward there, *i.e.*, $\rho(\perp) = \pm 1$. With this explicit construction of the environments, we can give an analytic lower bound of $\epsilon_\star(\pi)$, which coincides with $\|\mathcal{R}_\pi(w_\pi^\sharp)\|_{\text{TV}}$. See Section B for the complete proof. $\qquad\square$

An immediate consequence of Theorem 4.1 is that it tells us when and how the minimax bias is positive. To see this, let $\mu_\pi^{\not\sharp} := \mu_\pi - \mu_\pi^\sharp$ be the projection residual of $\mu_\pi$.

**Corollary 4.2.** *We have $\epsilon_\star(\pi) = \|\Delta_\pi \mu_\pi^{\not\sharp}\|_{\mathrm{TV}}$ and thus*

$$\|\mu_\pi^{\not\sharp}\|_{\mathrm{TV}} \leq \epsilon_\star(\pi) \leq \frac{1+\gamma}{1-\gamma} \|\mu_\pi^{\not\sharp}\|_{\mathrm{TV}}.$$

In other words, the minimax bias is zero if and only if the projection residual $\mu_\pi^{\not\sharp}$ is zero, or equivalently, if $\mu_\pi$ is absolutely continuous with respect to the data distribution $\beta$. Moreover, the size of the minimax bias is proportional to the size of the projection residual $\mu_\pi^{\not\sharp}$. It thus formally asserts the limitation of the point-based estimators in the insufficient exploration settings.

In summary, any OPE methods must be biased in the worst case whenever the exploration is insufficient, $\mu_\pi \not\ll \beta$, motivating the interval-based approach.

## 5   PROBLEM SETUP: MINIMAX-BIAS OPI

As discussed above, the point-based estimator suffers from irreducible bias, suggesting a hardness in Problem 3.1 under realistic assumptions. Given such a limitation, we propose an alternative problem setting called the *minimax-bias OPI* so that we can solve it under realistic assumptions. Since there exists a bias, our idea is to estimate the value by an *interval* that contains the true value, instead of a *point*.

Let us first define the target of the estimation, which we call the *optimal interval*. As discussed earlier, since $J_\star(\pi)$ and $\epsilon_\star(\pi)$ are the best possible point estimator and its error guarantee, respectively, the optimal interval can be naturally formulated as follows.

**Definition 5.1** (Optimal interval). *The following is referred to as the optimal interval:*

$$\mathcal{I}_\star(\pi) \equiv \mathcal{I}_\star(\pi; \mathcal{P}) := [J_\star(\pi) - \epsilon_\star(\pi), J_\star(\pi) + \epsilon_\star(\pi)].$$

Then, the problem of offline policy interval estimation (OPI), an uncertainty-aware interval extension of Problem 3.1, is formalized as follows.

**Problem 5.1** (Minimax-bias OPI). *Estimate $\mathcal{I}_\star(\pi)$ based on $(\mathcal{D}, \pi, \gamma)$, where $\mathcal{D} \sim G_{\mathcal{M}, \beta}$.*

Towards estimating the optimal interval, let us introduce two desirable properties of an interval. Definition 5.2 is stronger than Definition 5.3.

**Definition 5.2** (Approximate optimality). *We refer to the interval $\mathcal{I}$ satisfying $d_H(\mathcal{I}, \mathcal{I}_\star(\pi)) \leq \epsilon$, where $d_H(\cdot, \cdot)$ is the Hausdorff distance, as $\epsilon$-approximately optimal. Moreover, a sequence of intervals $\{\mathcal{I}_n\}_{n \geq 1}$ is said to be asymptotically (approximately) optimal if it converges to an (approximately) optimal interval.*

**Definition 5.3** (Validity). *An interval $\mathcal{I} \subset \mathbb{R}$ is said to be valid if $\mathcal{I} \supset \mathcal{I}_\star(\pi)$. Moreover, a sequence of intervals $\{\mathcal{I}_n\}_{n \geq 1}$ is said to be asymptotically valid if its lower limit $\lim_{k \to \infty} \bigcap_{n \geq k} \mathcal{I}_n$ is valid.*

## 6   THEORETICAL FOUNDATION OF MINIMAX-BIAS OPI

We provide a theoretical foundation to solve Problem 5.1, mainly referring to Theorem 4.1.

First, Theorem 4.1 implies that the MIS estimator $\hat{J}(w)$ with the projected importance weight $w = w_\pi^\sharp$ is optimal in the sense it achieves the minimax bias. More generally:

**Corollary 6.1.** *There exists $w \in L^1(\beta)$ such that $\epsilon[\hat{J}(w)] = \epsilon_\star(\pi)$.*

This motivates us to seek for the best weight function $w$ within the MIS framework. The following corollary shows the next significant implications, that the optimal point-based and interval-based OPE is achieved with combining the MIS estimator and the minimization of DBR.

**Corollary 6.2.** *Let $w_\star$ be a minimizer of DBR in a compact hypothesis class $\mathcal{W} \subset \mathscr{B}(\mathcal{X})$,*

$$w_\star \in \operatorname*{argmin}_{w \in \mathcal{W}} \|\mathcal{R}_\pi(w)\|_{\mathrm{TV}}. \tag{10}$$

*Then, we have*

$$\left| \mathbb{E}\hat{J}(w_\star) - J_\star(\pi) \right| \le \frac{1+\gamma}{1-\gamma}\epsilon_{\mathcal{W}}, \tag{11}$$

*and*

$$\left| \|\mathcal{R}_\pi(w_\star)\|_{\mathrm{TV}} - \epsilon_\star(\pi) \right| \le \frac{1+\gamma}{1-\gamma}\epsilon_{\mathcal{W}}, \tag{12}$$

*where $\epsilon_{\mathcal{W}} := \min_{w\in\mathcal{W}} \|w - w_\pi^\sharp\|_{L^1(\beta)}$ is the realizability error of $\mathcal{W}$.*

In other words, given $\mathcal{W}$ is expressive enough to approximate $w_\pi^\sharp$ well and thus $\epsilon_{\mathcal{W}}$ is negligible, the optimal point estimator $J_\star(\pi)$ and its uncertainty $\epsilon_\star(\pi)$ can be estimated with a solution $w_\star$ to (10) and its objective value $\|\mathcal{R}_\pi(w_\star)\|_{\mathrm{TV}}$, respectively. These observations naturally lead us to the following proxy to the optimal interval

$$\mathcal{I}(\pi; \mathcal{W}) := \left[ \mathbb{E}\hat{J}(w_\star) - \|\mathcal{R}_\pi(w_\star)\|_{\mathrm{TV}}, \ \mathbb{E}\hat{J}(w_\star) + \|\mathcal{R}_\pi(w_\star)\|_{\mathrm{TV}} \right]. \tag{13}$$

In fact, it satisfies two desirable properties: the validity and the approximate optimality.

**Theorem 6.3.** *The interval $\mathcal{I}(\pi; \mathcal{W})$ is valid and $2\frac{1+\gamma}{1-\gamma}\epsilon_{\mathcal{W}}$-approximately optimal.*

## 7 ESTIMATION OF OPTIMAL INTERVAL

As discussed in the previous section, the estimation of the optimal interval $\mathcal{I}_\star(\pi)$ is reduced to the minimization of the TV norm $\|\mathcal{R}_\pi(w)\|_{\mathrm{TV}}$. However, even the estimation of the exact TV norm is notorious for its difficulty (e.g., see Section 5 in Sriperumbudur et al. (2012)), let alone the minimization. This motivate us to develop new variational approximations for the TV norm. In particular, we newly introduce two approximations of $\|\mathcal{R}_\pi(w)\|_{\mathrm{TV}}$, each of which is suitable for the evaluation and the optimization of the objective, respectively.

### 7.1 EVALUATING THE OBJECTIVE

The first approximation reduces the evaluation of the TV norm to a conventional regression problem. To see this, let us begin with the approximation formula for the TV norm of general measures.

Let $\mathcal{F}$ be a *universal function approximator* on $\mathcal{X}$, *i.e.*, a set of functions dense in $\mathscr{C}(\mathcal{X})$, such as the reproducing kernel Hilbert space (RKHS) with a universal kernel (Sriperumbudur et al., 2010) or a set of neural networks (Hornik et al., 1989).

**Proposition 7.1.** *For all positive measures $P \in \mathscr{M}(\mathcal{X})$, we have*

$$\|P\|_{\mathrm{TV}} = \sup_{f\in\mathcal{F}} \langle \llbracket f \rrbracket, \ P \rangle \tag{14}$$

*where $\llbracket t \rrbracket := \max\{-1, \min\{1, t\}\}$ denotes the clipping of $t \in \mathbb{R}$ to $[-1, 1]$.*

The proof is relegated to Section C. Letting $P = \mathcal{R}_\pi(w)$, we immediately get the following special case useful for the evaluation of DBR.

**Corollary 7.2.** *For all $w \in \mathscr{B}(\mathcal{X})$, we have*

$$\|\mathcal{R}_\pi(w)\|_{\mathrm{TV}} = \sup_{f\in\mathcal{F}} \langle \llbracket f \rrbracket, \ \mathcal{R}_\pi(w) \rangle. \tag{15}$$

The supremum (15) is estimated via the regularized empirical risk minimization framework, minimizing the following objective

$$\hat{\mathcal{L}}(f) := -\langle \llbracket f \rrbracket, \ \hat{\mathcal{R}}_\pi(w) \rangle + \Psi(f), \tag{16}$$

where $\hat{\mathcal{R}}_\pi(w) := \hat{\iota}_\pi - (1-\gamma)^{-1}(I - \gamma\hat{T}_\pi)(w \odot \hat{\beta})$ is the natural empirical counterpart of the DBR, $\Psi : \mathcal{F} \to \mathbb{R}$ is a penalty function that makes it easier to minimize and prevent the minimizer from overfitting. Once the regularized empirical risk minimizer $\hat{f} := \arg\min_{f\in\mathcal{F}} \hat{\mathcal{L}}(f)$ is found, we then evaluate $\langle \llbracket \hat{f} \rrbracket, \ \hat{\mathcal{R}}_\pi(w) \rangle$ to approximate the RHS of (15) and get the desired estimate. The resultant procedure of the evaluation of the TV norm is summarized in Algorithm A.1.

In fact, under a reasonable choice on $\mathcal{F}$ and $\Psi$, it is shown that the output of Algorithm A.1 is consistent. The details on the specific choice of $\mathcal{F}$ and $\Psi$ and the proof is provided in Section D.

**Theorem 7.3.** *Let* `EvaluateDBR(D, F, w)` *be the output of Algorithm A.1, where $\mathcal{F}$ and $\Psi$ are given as in Section D.1. Then, for all $w \in \mathcal{W}$, we have* `EvaluateDBR(D, F, w)` $\to \|\mathcal{R}_\pi(w)\|_{\mathrm{TV}}$ *in probability.*

## 7.2 MINIMIZING THE OBJECTIVE

We now turn to the minimization of $\|\mathcal{R}_\pi(w)\|_{\mathrm{TV}}$ with respect to $w \in \mathcal{W}$. The previous variational formula (15) is not straightforwardly usable for this purpose since it ends up with the saddle-point problem, which we found is too unstable in practice. To mitigate this issue, we introduce a minimization-based approximation of the TV norm.

To this end, we first introduce the convolution of the TV norm with the maximum mean discrepancy (MMD) (Sriperumbudur et al., 2009). Here, the MMD of a measure $P \in \mathcal{M}(\mathcal{X})$ is given by $\mathrm{MMD}_\kappa(P) \coloneqq \sqrt{\langle \kappa, P^{\otimes 2} \rangle}$, where $\kappa : \mathcal{X}^2 \to \mathbb{R}$ is a $c_0$-universal kernel in the sense of Sriperumbudur et al. (2010) and $P^{\otimes 2}$ denotes the product measure of $P$ on $\mathcal{X}^2$.

**Definition 7.1** (Convolution norm). *For all $P \in \mathcal{M}(\mathcal{X})$ and $u \geq 1$, we refer to*

$$\|P\|_{u,\kappa} \coloneqq \inf_{Q \ll P} \{u \, \mathrm{MMD}_\kappa(P - Q) + \|Q\|_{\mathrm{TV}}\} \tag{17}$$

*as the $u$-convolution norm of $P$.*

The following proposition shows the $u$-convolution norm is a reasonable approximation of the TV norm and admits a sample-based estimation unlike the TV norm.

**Proposition 7.4.** *For all $P \in \mathcal{M}(\mathcal{X})$, we have*

$$\|P\|_{\mathrm{TV}} = \lim_{u \to \infty} \|P\|_{u,\kappa}. \tag{18}$$

*Moreover, if $P$ is a probability measure, for all $\delta \in (0,1)$, we have*

$$\|\hat{P}_n - P\|_{u,\kappa} = O\left(\sqrt{\frac{u^2 + \ln(1/\delta)}{n}}\right) \tag{19}$$

*with probability $\geq 1 - \delta$, where $\hat{P}_n \coloneqq \frac{1}{n}\sum_{i=1}^n \delta_{x_i}$ is the empirical distribution of an $n$-sample $(x_1, ..., x_n)$ independently drawn from $P$, $n \geq 1$.*

*Proof.* The key of the proof is Lemma E.2, which gives the dual representation of the convolution norm,

$$\|P\|_{u,\kappa} = \sup_{\substack{f \in \mathscr{B}(\mathcal{X}) \\ \|f\|_{\mathcal{H}} \leq u \\ \|f\|_\infty \leq 1}} \langle f, \, P \rangle,$$

where $\mathcal{H}$ is the RKHS generated by $\kappa$. Then, the density of the universal RKHS in $\mathscr{C}(\mathcal{X})$ implies that $\lim_{u \to \infty} \|P\|_{u,\kappa} = \sup_{\|f\|_\infty \leq 1} \langle f, \, P \rangle = \|P\|_{\mathrm{TV}}$, which proves the first statement. The second statement follows from the uniform law of large number, namely Theorem H.5 and Lemma H.6. $\quad\square$

Slightly extending it for the sample approximation of the DBR $\mathcal{R}_\pi(w) \approx \hat{\mathcal{R}}_\pi(w)$, we obtain the following approximation formula useful for the weight optimization. The proof is given in Section F.

**Corollary 7.5.** *For all $w \in \mathscr{B}(\mathcal{X})$, we have*

$$\|\hat{\mathcal{R}}_\pi(w)\|_{u,\kappa} \to \|\mathcal{R}_\pi(w)\|_{\mathrm{TV}} \tag{20}$$

*in probability as $u \to \infty$ and $n/u^2 \to \infty$.*

In the practical implementation, one may change the variable in the LHS of (20) from the measure $Q \in \mathcal{M}(\mathcal{X})$ to the function $g \in \mathscr{B}(\mathcal{X})$,

$$\begin{aligned}
\|\hat{\mathcal{R}}_\pi(w)\|_{u,\kappa} &= \inf_{Q \ll \hat{\mathcal{R}}_\pi(w)} \left\{u \, \mathrm{MMD}_\kappa(\hat{\mathcal{R}}_\pi(w) - Q) + \|Q\|_{\mathrm{TV}}\right\} \\
&= \min_{g \in \mathscr{B}(\mathcal{X})} \left\{u \, \mathrm{MMD}_\kappa(\hat{\mathcal{R}}_\pi(w) - g \odot \hat{\eta}_\pi) + \langle |g|, \, \hat{\eta}_\pi \rangle\right\},
\end{aligned} \tag{21}$$

---

**Algorithm 7.1:** Minimax Optimal Interval Estimation (MOI)

---

**Input:** Dataset $\mathcal{D}$, universal approximator $\mathcal{F}$, hypothesis class $\mathcal{W}$, kernel $\kappa$;
**Output:** Estimate of the optimal interval $\mathcal{I}_\star(\pi)$;

1 $\hat{w} := \texttt{OptimizeDBR}(\mathcal{D}, \mathcal{F}, \mathcal{W}, \kappa)$ ;                    // Algorithm A.2
2 $\hat{\epsilon} := \texttt{EvaluateDBR}(\mathcal{D}, \mathcal{F}, \hat{w})$ ;                    // Algorithm A.1
3 **return** $[\hat{J}(\hat{w}) - \hat{\epsilon}, J(\hat{w}) + \hat{\epsilon}]$;

---

where $\hat{\eta}_\pi := (\hat{\iota}_\pi + \hat{T}_\pi \hat{\beta} + \hat{\beta})/3$. Here, we have exploited the transitivity of the absolute continuity with $Q \ll \hat{\mathcal{R}}_\pi(w) \ll \hat{\eta}_\pi$. Note that the minimization with respect to $g$ is tractable since $g$ is only evaluated on the support of $\hat{\eta}_\pi$, which is a finite set. Moreover, the convolution-based formula (21) is stable with the minimization with respect to $w \in \mathcal{W}$ resulting in the joint minimization problem of $(w, g)$, unlike the regression-based formula (15) resulting in the saddle-point problem.

Since the objective of (21) is lower bounded and convex with respect to $(w, g)$, the minimizer

$$\hat{w}_u := \underset{w \in \mathcal{W}}{\arg\min} \|\hat{\mathcal{R}}_\pi(w)\|_{u,\kappa} \tag{22}$$

is computed with any convex optimization algorithms. The hyperparameter $u$ is then chosen from a predefined grid $\mathcal{U}$ so that the TV norm of DBR is minimized. Specifically, one may choose it as the logarithmically-even grid $\mathcal{U} := \{2^k\}_{k=0}^{k_{\max}}$, where the upper limit $k_{\max} := \lfloor \log_2 \sqrt{\min\{n, m\}} \rfloor$ is determined according to the order of empirical approximation error (19). The entire procedure of the weight estimation is summarized in Algorithm A.2. By its derivation, we can formally show the consistency of Algorithm A.2. The proof is relegated to Section G.

**Theorem 7.6.** *Let* $\texttt{OptimizeDBR}(\mathcal{D}, \mathcal{F}, \mathcal{W}, \kappa)$ *be the output of Algorithm A.2. Then,* $\texttt{OptimizeDBR}(\mathcal{D}, \mathcal{F}, \mathcal{W}, \kappa) \to w_\star$ *in probability as* $n \to \infty$.

Finally, we present our OPI method in Algorithm 7.1, called the *minimax optimal interval estimation (MOI)*, which is straightforwardly derived from Algorithm A.2 and the equation (13). We can also guarantee the validity and the approximate consistency of MOI in the asymptotic sense. The proof follows directly from Theorem 7.6.

**Theorem 7.7.** *Let* $\texttt{MOI}(\mathcal{D}, \mathcal{F}, \mathcal{W}, \kappa)$ *be the output of Algorithm 7.1. Then,* $\texttt{MOI}(\mathcal{D}, \mathcal{F}, \mathcal{W}, \kappa)$ *is asymptotically valid and* $2\frac{1+\gamma}{1-\gamma}\epsilon_{\mathcal{W}}$*-approximately optimal in probability.*

## 8 CONCLUSION

In this paper, we have studied OPI without the sufficient exploration and the realizability conditions. In particular, we have pointed out the existence of the irreducible bias in such a general setting, and correspondingly, introduced a novel formulation of the interval-based OPE. We have then revealed the connection between the conventional MIS estimator and the irreducible bias, which is eventually utilized to construct the proposed method, the minimax optimal interval estimator (MOI), and to prove its optimality.

One of the major limitations of the proposed method is its model agnosticity, lying at the opposite end to the model-based approach, e.g., Yu et al. (2020), that depends on the full correctness of the model. It is left for future work to extend and combine these methods to be applicable to partially correct models.

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

## CONTENTS

## A  ALGORITHMS

## B  PROOF OF THEOREM 4.1

We first show the first two inequality.

---

**Algorithm A.1:** Evaluation of DBR

---

**Input:** Dataset $\mathcal{D}$, function approximator $\mathcal{F}$, density-ratio estimate $w$;
**Output:** Estimate of $\|\mathcal{R}_\pi(w)\|_{\mathrm{TV}}$;

1   $\hat{f} \leftarrow \operatorname{argmin}_{f \in \mathcal{F}} \hat{\mathcal{L}}(f)$, where $\hat{\mathcal{L}}(f)$ is given by (16) ;      `// See also Section D`
2   $\hat{b} \leftarrow \langle [\![\hat{f}]\!],\ \hat{\mathcal{R}}_\pi(w) \rangle$;
3   **return** $\hat{b}$;

---

**Algorithm A.2:** Optimization of DBR

---

**Input:** Dataset $\mathcal{D}$, universal approximator $\mathcal{F}$, hypothesis class $\mathcal{W}$, kernel $\kappa$;
**Output:** Approximate minimizer of $\|\mathcal{R}_\pi(w)\|_{\mathrm{TV}}$;

1   **for** $u \in \mathcal{U}$ **do**
2     $\hat{w}_u := \operatorname{argmin}_{w \in \mathcal{W}} \|\hat{\mathcal{R}}_\pi(w)\|_{u,\kappa}$;
3     $\hat{\epsilon}_u := \texttt{EvaluateDBR}(\mathcal{D}, \mathcal{F}, \hat{w}_u)$ ;      `// Algorithm A.1`
4   $\hat{u} := \operatorname{argmin}_{u \in \mathcal{U}} \hat{\epsilon}_u$;
5   **return** $\hat{w}_{\hat{u}}$;

---

**Lemma B.1.** *For all $w \in \mathscr{B}(\mathcal{X})$,*

$$\epsilon_\star(\pi) \le \epsilon\left[\hat{J}(w)\right] \le \|\mathcal{R}_\pi(w)\|_{\mathrm{TV}} \tag{23}$$

*Proof.* The first inequality is trivial. To show the second one, observe

$$
\begin{aligned}
\left| J_\mathcal{M}(\pi) - \mathbb{E}\hat{J}(w) \right| &= |\langle \rho,\ \Gamma_\pi \iota_\pi - w \odot \beta \rangle| \\
&= |\langle q_\pi,\ \iota_\pi - \Delta_\pi(w \odot \beta) \rangle| \\
&= |\langle q_\pi,\ \mathcal{R}_\pi(w) \rangle| \\
&\le \|\mathcal{R}_\pi(w)\|_{\mathrm{TV}},
\end{aligned}
$$

where the last inequality is owing to $\|q_\pi\|_\infty \le 1$. Since the RHS is independent of $\mathcal{M}$ given $G_{\mathcal{M},\beta}$, we thus have

$$\epsilon\left[\hat{J}(w)\right] = \sup_{\mathcal{M}': G_{\mathcal{M}',\beta} = G_{\mathcal{M},\beta}} \left| J_{\mathcal{M}'}(\pi) - \mathbb{E}\hat{J}(w) \right| \le \|\mathcal{R}_\pi(w)\|_{\mathrm{TV}}. \tag{24}$$

$\square$

Now, to prove the last inequality, we prepare two extreme, yet indistinguishable environments $\mathcal{M}_\pm := (\iota, \tilde{T}, R_\pm)$.

Let $\tilde{T}$ be an arbitrary state-transition operator indistinguishable from $T$, which will be determined later. Also let $\tilde{T}_\pi$ be the state-action transition operator associated with $\tilde{T}$ and $\pi$ such that $\mathrm{d}\tilde{T}_\pi(s,a|x) = \mathrm{d}\tilde{T}(s|x)\mathrm{d}\pi(a|s)$ for $x, (s,a) \in \mathcal{X}$. Let $\tilde{\mu}_\pi := (1-\gamma)(I - \gamma\tilde{T}_\pi)^{-1}\iota_\pi$ be the common occupancy measure of $\mathcal{M}_\pm$ induced with $\tilde{T}$, and $\tilde{\mu}_\pi|_{\not\ll\beta}$ be the singular component of $\tilde{\mu}_\pi$ with respect to $\beta$. Let $\mathcal{X}_0$ be a set separating $\tilde{\mu}_\pi|_{\not\ll\beta}$ from $\beta$ and $\mathcal{X}_\beta := \mathcal{X} \setminus \mathcal{X}_0$ be its complement.[3] For convenience, let $\Pi_\beta, \Pi_0 : \mathscr{M}(\mathcal{X}) \to \mathscr{M}(\mathcal{X})$ denote the projections of measure onto $\mathcal{X}_\beta$ and $\mathcal{X}_0$, respectively, given by $\Pi_\beta := \chi_\beta \odot P$, and $\Pi_0 P := (1 - \chi_\beta) \odot P$ where $\chi_\beta$ being the indicator function of $\mathcal{X}_\beta$ such that $\chi_\beta(x) = 1$ if $x \in \mathcal{X}_\beta$ and $\chi_0(x) = 0$ otherwise. Note that $\tilde{\mu}_\pi|_{\not\ll\beta} = \Pi_0 \tilde{\mu}_\pi$ by construction.

Finally, put $R_\pm(x) = \delta_{\pm 1}$ for $x \in \mathcal{X}_0$ and $R_\pm(x) = R(x)$ otherwise, which is necessary for the indistinguishability of $R_\pm$, and denote the associated expected reward by $\rho_\pm(x) := \int r \mathrm{d}R_\pm(r|x)$, $x \in \mathcal{X}$. Then, we have

$$J_+(\pi) - J_-(\pi) = \langle \rho_+ - \rho_-, \tilde{\mu}_\pi \rangle = 2\|\tilde{\mu}_\pi|_{\not\ll\beta}\|_{\mathrm{TV}}, \tag{25}$$

---

[3]That is, $\{\mathcal{X}_0, \mathcal{X}_\beta\}$ is a partition of $\mathcal{X}$ such that $\tilde{\mu}_\pi|_{\not\ll\beta}(E) = 0$ for all measurable $E \subset \mathcal{X}_\beta$ and $\beta(E) = 0$ for all measurable $E \subset \mathcal{X}_0$.

where $J_{\pm}(\pi)$ are the policy values with respect to $\mathcal{M}_{\pm}$.

Now, the following lemma connects the RHS of (25) with DBR.

**Lemma B.2.** *There exist $\tilde{T}$ indistinguishable from $T$ such that*

$$\|\tilde{\mu}_\pi|_{\not\ll\beta}\|_{\mathrm{TV}} = \|\mathcal{R}_\pi(w_\pi^\sharp)\|_{\mathrm{TV}}. \tag{26}$$

*Proof.* The proof is constructive. Consider an expanded state space $\mathcal{S} \leftarrow \mathcal{S} \cup \{\perp\}$, where $\perp$ denotes an absorbing state of $\tilde{T}$. Accordingly, let $\mathcal{X}_0 \leftarrow \mathcal{X}_0 \cup (\{\perp\} \times \mathcal{A})$. Now, put $\tilde{T} := \tilde{T}|_{\ll\beta} + \tilde{T}|_{\not\ll\beta}$ such that $\tilde{T}|_{\ll\beta}$ is the restriction of $T$ onto $\mathcal{X}_\beta$ and $\tilde{T}|_{\not\ll\beta}$ is the absorbing transition, respectively given by

$$\tilde{T}|_{\ll\beta} = T\Pi_\beta, \qquad \tilde{T}|_{\not\ll\beta}P = P(\mathcal{X}_0)\,\delta_\perp, \qquad P \in \mathscr{M}(\mathcal{X}).$$

Here, $\tilde{T}|_{\ll\beta}$ is corresponding to the known component of $\tilde{T}$, which necessarily coincides with that of $T$ by definition, and $\tilde{T}|_{\not\ll\beta}$ is corresponding to the unknown component of $\tilde{T}$. Note that $\tilde{T}$ is a proper transition operator indistinguishable from $T$.

Let $\tilde{T}_\pi|_{\ll\beta}$ and $\tilde{T}_\pi|_{\not\ll\beta}$ be the state-action transition operator associated with $T_\beta$ and $T_0$, respectively, such that $\mathrm{d}\tilde{T}_\pi|_{\ll\beta}(s,a|x) := \mathrm{d}\tilde{T}|_{\ll\beta}(s|x)\mathrm{d}\pi(a|s)$ and $\mathrm{d}\tilde{T}_\pi|_{\not\ll\beta}(s,a|x) := \mathrm{d}\tilde{T}|_{\not\ll\beta}(s|x)\mathrm{d}\pi(a|s)$ for $x,(s,a) \in \mathcal{X}$. Then, we have

$$
\begin{aligned}
(1-\gamma)\iota_\pi &= (I - \gamma\tilde{T}_\pi)\tilde{\mu}_\pi \\
&= (I - \gamma\tilde{T}_\pi|_{\ll\beta} - \gamma\tilde{T}_\pi|_{\not\ll\beta})\tilde{\mu}_\pi \\
&= (I - \gamma\tilde{T}_\pi|_{\ll\beta})\tilde{\mu}_\pi - \gamma\tilde{\mu}_\pi(\mathcal{X}_0)\delta_{\perp,\pi} \\
&= (I - \gamma\tilde{T}_\pi|_{\ll\beta})\tilde{\mu}_\pi - \gamma\|\tilde{\mu}_\pi|_{\not\ll\beta}\|_{\mathrm{TV}}\delta_{\perp,\pi},
\end{aligned}
$$

which implies, with $\tilde{P} := (I - \gamma\tilde{T}_\pi|_{\ll\beta})^{-1}\iota_\pi$,

$$
\begin{aligned}
\tilde{\mu}_\pi &= (1-\gamma)\tilde{P} + \gamma\|\tilde{\mu}_\pi|_{\not\ll\beta}\|_{\mathrm{TV}}(I - \gamma\tilde{T}_\pi|_{\ll\beta})^{-1}\delta_{\perp,\pi} \\
&= (1-\gamma)\tilde{P} + \gamma\|\tilde{\mu}_\pi|_{\not\ll\beta}\|_{\mathrm{TV}}\delta_{\perp,\pi}. && (\because \tilde{T}_\pi|_{\ll\beta}\delta_{\perp,\pi} = 0)
\end{aligned}
$$

Measuring the volumes on $\mathcal{X}_0$, we further get

$$\|\tilde{\mu}_\pi|_{\not\ll\beta}\|_{\mathrm{TV}} = (1-\gamma)\tilde{P}(\mathcal{X}_0) + \gamma\|\tilde{\mu}_\pi|_{\not\ll\beta}\|_{\mathrm{TV}},$$

which yields

$$\tilde{P}(\mathcal{X}_0) = \|\tilde{\mu}_\pi|_{\not\ll\beta}\|_{\mathrm{TV}}. \tag{27}$$

On the other hand, since $w_\pi^\sharp = (1-\gamma)\mathrm{d}(\Pi_\beta\tilde{P})/\mathrm{d}\beta$, we have

$$
\begin{aligned}
\|\mathcal{R}_\pi(w_\pi^\sharp)\|_{\mathrm{TV}} &= \left\|\iota_\pi - (1-\gamma)\Delta_\pi\Pi_\beta\tilde{P}\right\|_{\mathrm{TV}} \\
&= \left\|\iota_\pi - (\Pi_\beta - \gamma\tilde{T}_\pi|_{\ll\beta})\tilde{P}\right\|_{\mathrm{TV}} && \left(\because T_\pi\Pi_\beta = \tilde{T}_\pi|_{\ll\beta}\right) \\
&= \left\|\iota_\pi - \Pi_\beta\iota_\pi + \gamma\Pi_0 T_\pi|_{\ll\beta}\tilde{P}\right\|_{\mathrm{TV}} && \left(\because \tilde{T}_\pi|_{\ll\beta} = \Pi_\beta\tilde{T}_\pi|_{\ll\beta} + \Pi_0\tilde{T}_\pi|_{\ll\beta}\right) \\
&= \left\|\Pi_0\left(\iota_\pi + \gamma T_\pi|_{\ll\beta}\tilde{P}\right)\right\|_{\mathrm{TV}} \\
&= \left\|\Pi_0\tilde{P}\right\|_{\mathrm{TV}} && \left(\because (I - \gamma\tilde{T}_\pi|_{\ll\beta})\tilde{P} = \iota_\pi\right) \\
&= \tilde{P}(\mathcal{X}_0).
\end{aligned}
$$

Combining it with (27), we get the desired result. $\qquad\square$

Plugging (26) into (25), we have

$$\|\mathcal{R}_\pi(w_\pi^\sharp)\|_{\mathrm{TV}} = \frac{1}{2}\left\{J_+(\pi) - J_-(\pi)\right\} \tag{28}$$

with a specific configuration of $\tilde{T}$. Since $\mathcal{M}_\pm$ are indistinguishable from one another, any estimators must incur the bias of at least a half of the difference $J_+(\pi) - J_-(\pi)$ in the worst case, *i.e.*, $\mathcal{R}_\pi(w_\pi^\sharp) \leq \epsilon_\star(\pi)$. This proves the last inequality of (8) via the triangle inequality,

$$\|\mathcal{R}_\pi(w)\|_{\mathrm{TV}} \leq \|\mathcal{R}_\pi(w_\pi^\sharp)\|_{\mathrm{TV}} + \|\Delta_\pi(w - w_\pi^\sharp) \odot \beta\|_{\mathrm{TV}}$$

$$\leq \epsilon_\star(\pi) + \frac{1+\gamma}{1-\gamma}\|w - w_\pi^\sharp\|_{L^1(\beta)}.$$

and thus concludes the proof of Proposition 4.1.

## C    PROOF OF PROPOSITION 7.1

First, we introduce a saddle-point formulation of the TV norm. Let $\mathcal{F}_1 \coloneqq \{f \in \mathcal{F} : \|f\|_\infty \leq 1\}$ be the intersection of $\mathcal{F}$ with the unit ball of $\mathscr{B}(\mathcal{X})$. Lemma C.1 gives a general variational formula of the TV norm.

**Lemma C.1.** *For all $P \in \mathscr{M}(\mathcal{X})$, we have*

$$\|P\|_{\mathrm{TV}} = \sup_{f \in \mathcal{F}_1} \langle f, P \rangle. \tag{29}$$

*Proof.* Let us denote the unit ball of $\mathscr{B}(\mathcal{X})$ with $\mathcal{U}_1 \coloneqq \{f \in \mathscr{B}(\mathcal{X}) : \|f\|_\infty \leq 1\}$. As an instance of the integral probability metrics (IPM), the TV norm is known to be written as

$$\|P\|_{\mathrm{TV}} = \sup_{g \in \mathcal{U}_1} \langle g, P \rangle. \tag{30}$$

Now fix $g \in \mathcal{U}_1$ such that $|\|P\|_{\mathrm{TV}} - \langle g, P \rangle| \leq c$ for a positive constant $c > 0$. Since $\mathcal{F}_1$ is dense in $\mathscr{C}(\mathcal{X}) \cap \mathcal{U}_1$, it is also dense in $L^1(P) \cap \mathcal{U}_1$ and therefore there exists $f \in \mathcal{F}_1$ such that $\|f - g\|_{L^1(P)} \leq c$. Then it follows that

$$|\|P\|_{\mathrm{TV}} - \langle f, P \rangle| \leq |\|P\|_{\mathrm{TV}} - \langle g, P \rangle| + \|f - g\|_{L^1(P)} \leq 2c. \tag{31}$$

Since $c > 0$ can be arbitrarily small, we finally have

$$\|P\|_{\mathrm{TV}} \leq \sup_{f \in \mathcal{F}_1} \langle f, P \rangle \leq \sup_{g \in \mathcal{U}_1} \langle g, P \rangle \leq \|P\|_{\mathrm{TV}}, \tag{32}$$

which proves the desired result. $\qquad\square$

Since the supremum in (29) is taken over a constrained domain $\mathcal{F}_1$, its computation is not necessarily tractable in general. The following lemma is useful to make the domain unconstrained.

**Lemma C.2.** *Let $\Psi : \mathscr{B}(\mathcal{X}) \to \mathbb{R}_{\geq 0}$ be a penalty function such that $\Psi(f) = 0$ if $f \in \mathcal{F}_1$. Then, for all $P \in \mathscr{M}(\mathcal{X})$, we have*

$$\sup_{f \in \mathcal{F}_1} \langle f, P \rangle = \sup_{f \in \mathcal{F}} \{\langle [\![f]\!], P \rangle - \Psi(f)\}, \tag{33}$$

*where $[\![f]\!](x) \coloneqq \max\{-1, \min\{1, f(x)\}\}$ denotes the clipping of $f(x)$ to $[-1, 1]$.*

*Proof.* Slightly extending (32), we have

$$\begin{aligned}
\|P\|_{\mathrm{TV}} &\leq \sup_{f \in \mathcal{F}_1} \langle f, P \rangle \\
&= \sup_{f \in \mathcal{F}_1} \{\langle f, P \rangle - \Psi(f)\} \\
&\leq \sup_{f \in \mathcal{F}} \{\langle [\![f]\!], P \rangle - \Psi(f)\} && (\because \text{domain expansion}) \\
&\leq \sup_{g \in \mathcal{U}_1} \langle g, P \rangle && (\because [\![f]\!] \in \mathcal{U}_1, \Psi(f) \geq 0) \\
&\leq \|P\|_{\mathrm{TV}}.
\end{aligned}$$

This yields the desired claim. $\qquad\square$

Finally, Proposition 7.1 is proved taking the penalty function as the trivial one, $\Psi(f) = 0$ for all $f \in \mathcal{F}$. We utilize a nontrivial penalty function in Section D.

# D DETAILS AND PROOF OF THEOREM 7.3

We first present a preferred choice of the function approximator $\mathcal{F}$ and the penalty function $\Psi$, which is needed to construct the objective function $\hat{\mathcal{L}}(f)$ in (16). Then, we prove Theorem 7.3 to show the consistency of the resultant algorithm (Algorithm A.1).

## D.1 CHOICE OF FUNCTION APPROXIMATOR AND PENALTY FUNCTION

As for the function approximator $\mathcal{F}$, we choose a universal RKHS. Let $\kappa : \mathcal{X}^2 \to \mathbb{R}$ be the corresponding symmetric positive-definite kernel. We assume $\kappa$ is $c_0$-universal in the sense of Sriperumbudur et al. (2010). Also, without loss of generality, we assume $\kappa$ is normalized, $\|\kappa\|_\infty :=$ $\sup_{x,x' \in \mathcal{X}} |\kappa(x,x')| \leq 1$. For instance, the Gaussian kernel $\kappa(x,y) = \exp\{-\|x-y\|_2^2/(2\alpha^2)\}$, $x, y \in \mathbb{R}^d$, $d \geq 1$, $\alpha > 0$, is one of such choices.

As for the penalty function $\Psi$, we employ

$$\hat{\Psi}_\lambda(f) := \langle (|f| - 1)_+, \ \hat{\mathcal{R}}_{\pi,+}(w) + \hat{\mathcal{R}}_{\pi,-}(w) \rangle + \frac{\lambda}{2(1-\gamma)} \|f\|_{\mathcal{F}}^2, \tag{34}$$

where $\lambda > 0$ is a hyperparameter, $(g)_+ := \max\{0, g\}$ denotes the positive part of a function $g \in \mathcal{B}(\mathcal{X})$, $\hat{\mathcal{R}}_{\pi,+}(w) := \hat{\iota}_\pi - \frac{\gamma}{1-\gamma} \hat{T}_\pi(w \odot \hat{\beta})$ and $\hat{\mathcal{R}}_{\pi,-}(w) := \frac{1}{1-\gamma}(w \odot \hat{\beta})$ are the positive and the negative part of the empirical DBR, respectively, and $\| \cdot \|_{\mathcal{F}}$ is the RKHS norm. Then, letting $C(P) := 1 + \frac{1+\gamma}{1-\gamma} \langle w, \ P \rangle$, $P \in \mathcal{M}(\mathcal{X})$, the penalized objective (16) is simplified as

$$\hat{\mathcal{L}}(f) = \langle |f - 1|, \ \hat{\mathcal{R}}_{\pi,+}(w) \rangle + \langle |f + 1|, \ \hat{\mathcal{R}}_{\pi,-}(w) \rangle + \frac{\lambda}{2(1-\gamma)} \|f\|_{\mathcal{F}}^2 - C(\hat{\beta}), \tag{35}$$

which is convex with respect to $f \in \mathcal{F}$. In other words, the minimizer

$$\hat{f} \equiv \hat{f}_\lambda := \underset{f \in \mathcal{F}}{\operatorname{argmin}} \ \hat{\mathcal{L}}(f) \tag{36}$$

can be found in a tractable manner with convex optimization methods. As for the choice of $\lambda$, as will be seen in the next section, we can achieve the consistency if $\lambda \to 0$ and $n\lambda \to \infty$. Thus, we may employ some fixed default $\lambda = 1/\sqrt{n}$ or select the best hyperparameter within some fixed grid, *e.g.*, $\Lambda_n := \{1, 2, ..., 2^{\lfloor \log_2 n \rfloor}\}$, that best attains the supremum (15), possibly using the training-validation split technique.

## D.2 CONSISTENCY ANALYSIS

We first introduce some notations useful for the analysis. Let us define probability measures $P, \hat{P}_n \in \mathcal{M}(\mathcal{X}^3)$ by

$$dP(x_1, x_2, x_3) := d\iota_\pi(x_1) d\beta(x_2) dT_\pi(x_3|x_2),$$

$$d\hat{P}_n(x_1, x_2, x_3) := d\hat{\iota}_\pi(x_1) d\hat{\beta}(x_2) d\hat{T}_\pi(x_3|x_2),$$

loss functions $\ell_f, \varphi_f : \mathcal{X}^3 \to \mathbb{R}$, $f \in \mathcal{F}$, by

$$\ell_f(x_1, x_2, x_3) := \sum_{j=1}^{3} \ell_{f,j}(x_1, x_2, x_3), \qquad \varphi_f(x_1, x_2, x_3) := \sum_{j=1}^{3} \varphi_{f,j}(x_1, x_2, x_3),$$

for $x_1, x_2, x_3 \in \mathcal{X}$, where

$$\begin{aligned} \ell_{f,1}(x_1, x_2, x_3) &:= (1 - \gamma)|f(x_1) - 1|, & \varphi_{f,1}(x_1, x_2, x_3) &:= -(1 - \gamma)[\![f(x_1)]\!], \\ \ell_{f,2}(x_1, x_2, x_3) &:= |w(x_2)f(x_2) + |w(x_2)|\!|, & \varphi_{f,2}(x_1, x_2, x_3) &:= w(x_2)[\![f(x_2)]\!], \\ \ell_{f,3}(x_1, x_2, x_3) &:= \gamma|w(x_2)f(x_3) - |w(x_2)|\!|, & \varphi_{f,3}(x_1, x_2, x_3) &:= -\gamma w(x_2)[\![f(x_3)]\!]. \end{aligned}$$

Let us also define the associated risk functions

$$\mathcal{L}_\lambda(f; Q) := \langle \ell_f, \ Q \rangle + \frac{\lambda}{2} \|f\|_{\mathcal{F}}^2, \qquad\qquad \Phi(f; Q) := \langle \varphi_f, \ Q \rangle,$$

for probability measures $Q \in \mathscr{M}(\mathcal{X}^3)$. By these definitions, we have

$$\Phi(f; P) = (1 - \gamma)\langle [\![f]\!], \ \mathcal{R}_\pi(w)\rangle,$$
$$\Phi(f; \hat{P}_n) = (1 - \gamma)\langle [\![f]\!], \ \hat{\mathcal{R}}_\pi(w)\rangle,$$

and

$$\mathcal{L}_\lambda(f; P) = \Phi(f; P) + (1 - \gamma)\{\Psi_\lambda(f) + C(\beta)\} \tag{37}$$

$$\mathcal{L}_\lambda(f; \hat{P}_n) = \Phi(f; \hat{P}_n) + (1 - \gamma)\left\{\hat{\Psi}_\lambda(f) + C(\hat{\beta})\right\} \tag{38}$$

for all $\lambda > 0$ and $f \in \mathcal{F}$, where $\hat{\Phi}_\lambda$ is given by (34) and

$$\Psi_\lambda(f) := \langle(|f| - 1)_+, \ \mathcal{R}_{\pi,+}(w) + \mathcal{R}_{\pi,-}(w)\rangle + \frac{\lambda}{2(1 - \gamma)}\|f\|_{\mathcal{F}}^2, \tag{39}$$

$\mathcal{R}_{\pi,+}(w) := \iota_\pi + \frac{\gamma}{1-\gamma}T_\pi(w \odot \beta)$ and $\mathcal{R}_{\pi,-}(w) := \frac{1}{1-\gamma}(w \odot \beta)$ are the positive and the negative parts of the DBR, respectively.

Therefore, we obtain an alternative expression of the objective (16)

$$\hat{\mathcal{L}}(f) = \frac{1}{1 - \gamma}\mathcal{L}_\lambda(f; \hat{P}_n) - C(\hat{\beta}),$$

which implies

$$\hat{f}_\lambda = \underset{f \in \mathcal{F}}{\operatorname{argmin}} \, \mathcal{L}_\lambda(f; \hat{P}_n). \tag{40}$$

Moreover, by Lemma C.2, we also obtain alternative expressions of the quantity of interest

$$\|\mathcal{R}_\pi(w)\|_{\mathrm{TV}} = -\frac{1}{1 - \gamma} \inf_{f \in \mathcal{F}} \Phi(f; P) \tag{41}$$

$$= C(\beta) - \frac{1}{1 - \gamma} \inf_{f \in \mathcal{F}} \mathcal{L}_0(f; P). \tag{42}$$

The goal of this section is to reveal the relationship of $\hat{f}_\lambda$ and $\|\mathcal{R}_\pi(w)\|_{\mathrm{TV}}$ via (40), (41) and (42)

The following lemma gives a key insight on the behavior of $\hat{f}_\lambda$. Let $G := 1 - \gamma + (1 + \gamma)\|w\|_\infty$ be the Lipschitz constant of $f \mapsto \ell_f$ and $f \mapsto \varphi_f$. Let $\mathcal{B}_\mathcal{F}(0, 1) := \{f \in \mathcal{F} : \|f\|_\mathcal{F} \le 1\}$ be the unit closed ball of $\mathcal{F}$. Let $\mathfrak{R}_n(\mathcal{H})$ is the Rademacher complexity of a function class $\mathcal{H} \subset \mathscr{B}(\mathcal{X})$ (see Definition H.4).

**Lemma D.1.** *Suppose the predictor attaining $\min_{f \in \mathcal{F}} \mathcal{L}_\lambda(f; P)$ exists and denote it by $f_\lambda^* \in \mathcal{F}$. Also suppose $\|f\|_\infty \le \|f\|_\mathcal{F}$ for all $f \in \mathcal{F}$. Then, for all $\delta \in (0, 1)$, we have*

$$\mathcal{L}_\lambda(\hat{f}_\lambda; P) \le \mathcal{L}_\lambda(f_\lambda^*; P) + \frac{8G^2}{\lambda}\left\{\mathfrak{R}_n(\mathcal{B}_\mathcal{F}(0, 1)) + \sqrt{\frac{\ln(1/\delta)}{2n}}\right\}^2$$

*and*

$$\|\hat{f}_\lambda - f_\lambda^*\|_\mathcal{F} \le \frac{4G}{\lambda}\left\{\mathfrak{R}_n(\mathcal{B}_\mathcal{F}(0, 1)) + \sqrt{\frac{\ln(1/\delta)}{2n}}\right\}$$

*with probability $1 - \delta$.*

*Proof.* Define

$$\epsilon(f; Q) := \mathcal{L}_\lambda(f; Q) - \mathcal{L}_\lambda(f_\lambda^*; Q)$$

for $f \in \mathcal{F}$ and $Q \in \mathscr{M}(\mathcal{X})$. Let $\mathcal{F}_c := \{f \in \mathcal{F} : \epsilon(f; P) \le c\}$ and let $\tilde{\ell}_{g,j} := \ell_{f_\lambda^* + g, j} - \ell_{f_\lambda^*, j}$ for $j = 1, 2, 3$. Then, since $\epsilon(f; P) \ge \frac{\lambda}{2}\|f - f_\lambda^*\|_\mathcal{F}^2 \ge \frac{\lambda}{2}\|f - f_\lambda^*\|_\infty^2$ by the strong convexity of

$f \mapsto \mathcal{L}_\lambda(f; P)$, the uniform law of large number (Theorem H.5) gives

$$\sup_{f \in \mathcal{F}_c} \left\{ \epsilon(f; P) - \epsilon(f; \hat{P}_n) \right\}$$

$$= \sup_{f \in \mathcal{F}_c} \sum_{j=1}^3 \langle \tilde{\ell}_{f - f_\lambda^*, j}, \ P - \hat{P}_n \rangle$$

$$\leq \sup_{\|g\|_{\mathcal{F}} \leq \sqrt{2c/\lambda}} \sum_{j=1}^3 \langle \tilde{\ell}_{g,j}, \ P - \hat{P}_n \rangle \qquad (\because \text{strong convexity})$$

$$\leq 2 \mathfrak{R}_n \left( \left\{ \sum_{j=1}^3 \tilde{\ell}_{g,j} \ : \ \|g\|_{\mathcal{F}} \leq \sqrt{\frac{2c}{\lambda}} \right\} \right) + 2G \sqrt{\frac{2c}{\lambda}} \sqrt{\frac{\ln(1/\delta)}{2n}} \qquad (\because \text{Theorem H.5})$$

$$\leq \sqrt{\frac{8c}{\lambda}} G \left\{ \mathfrak{R}_n(\mathcal{B}_{\mathcal{F}}(0,1)) + \sqrt{\frac{\ln(1/\delta)}{2n}} \right\} =: \bar{\epsilon}(c, \delta) \qquad (\because \text{see below})$$

with probability $1 - \delta$ for all $c > 0$. Here, the last inequality follows from

$$\mathfrak{R}_n \left( \left\{ \sum_{j=1}^3 \tilde{\ell}_{g,j} \ : \ \|g\|_{\mathcal{F}} \leq \sqrt{\frac{2c}{\lambda}} \right\} \right) \leq \sum_{j=1}^3 \mathfrak{R}_n \left( \left\{ \tilde{\ell}_{g,j} \ : \ \|g\|_{\mathcal{F}} \leq \sqrt{\frac{2c}{\lambda}} \right\} \right) \qquad (\because \text{subadditivity})$$

$$\leq \sum_{j=1}^3 G_j \sqrt{\frac{2c}{\lambda}} \mathfrak{R}_n(\mathcal{B}_{\mathcal{F}}(0,1)) \qquad (\because \text{Lemma H.4})$$

$$\leq \sqrt{\frac{2c}{\lambda}} G \mathfrak{R}_n(\mathcal{B}_{\mathcal{F}}(0,1)),$$

where $G_j$ is the Lipschitz constants of $\ell_{f,j}$, $j = 1, 2, 3$. Now take $\tilde{f}_c \in \mathcal{F}$ such that $\tilde{f}_c = \hat{f}_\lambda$ if $\epsilon(\hat{f}_\lambda; P) \leq c$ and, otherwise, $\epsilon(\tilde{f}_c; \hat{P}_n) \leq 0$ and $\epsilon(\tilde{f}_c; P) = c$.[4] Then, when $c > \bar{\epsilon}(c, \delta)$, we get with probability $1 - \delta$

$$\epsilon(\tilde{f}_c; P) = \epsilon(\tilde{f}_c; \hat{P}_n) + \epsilon(\tilde{f}_c; P) - \epsilon(\tilde{f}_c; \hat{P}_n)$$

$$\leq \sup_{f: \epsilon(f; P) \leq c} \left\{ \epsilon(f; P) - \epsilon(f; \hat{P}_n) \right\} \qquad (\because \epsilon(\tilde{f}_c; \hat{P}_n) \leq 0, \ \epsilon(\tilde{f}_c; P) \leq c)$$

$$< c,$$

which implies $\tilde{f}_c = \hat{f}_\lambda$ and hence $\epsilon(\hat{f}_\lambda; P) < c$ with the same probability. Thus, since it holds with any $c > 0$ such that $c > \bar{\epsilon}(c, \delta)$, we finally have

$$\epsilon(\hat{f}_\lambda; P) \leq c^*$$

with probability $1 - \delta$, where $c^*$ is the solution to $c^* = \bar{\epsilon}(c^*, \delta)$, or more concretely

$$c^* = \frac{8G^2}{\lambda} \left\{ \mathfrak{R}_n(\mathcal{B}_{\mathcal{F}}(0,1)) + \sqrt{\frac{\ln(1/\delta)}{2n}} \right\}^2.$$

This concludes the proof. $\qquad \square$

Now, verifying the assumptions of Lemma D.1 and evaluating the Rademacher complexity of the unit ball $\mathfrak{R}_n(\mathcal{B}_{\mathcal{F}}(0,1))$, we get the following proposition.

**Proposition D.2** (Generalization error bound of RERM with RKHS). *For all $\delta \in (0,1)$ and $\lambda > 0$, we have*

$$\mathcal{L}_0(\hat{f}_\lambda; P) \leq \inf_{f \in \mathcal{H}} \mathcal{L}_\lambda(f; P) + \frac{8G^2 \ln(e^2/\delta)}{\lambda n}$$

---

[4]Such an $\tilde{f}_c$ exists at the intersection of the line segment $[f_\lambda^*, \hat{f}_\lambda]$ and the level set $\{f \in \mathcal{F} \ : \ \epsilon(f; P) \leq c\}$ since $f \mapsto \epsilon(f; Q)$ is convex.

*and*

$$\|\hat{f}_\lambda - f_\lambda^*\|_\mathcal{F} \leq \frac{4G}{\lambda}\sqrt{\frac{\ln(e^2/\delta)}{2n}} \tag{43}$$

*with probability* $1 - \delta$.

*Proof.* It suffices to invoke Lemma D.1 with Lemma H.6. To this end, we need to verify the existence of $\min_{f\in\mathcal{H}} \mathcal{L}_\lambda(f; P)$ and the dominance of the norm $\|\cdot\|_\infty \leq \|\cdot\|_\mathcal{H}$. In fact, the minimum exists since $f \mapsto \mathcal{L}_\lambda(f; P)$ is continuous with respect to $L^2(P)$ and the infimum $\inf_{f\in\mathcal{H}} \mathcal{L}_\lambda(f; P)$ does not change if we restrict the domain to the ball $\{f \in \mathcal{H} \ : \ \|f\|_\mathcal{H} \leq \sqrt{2\mathcal{L}_\lambda(0; P)}\}$, which is compact according to Lemma H.7. The dominance of the norm is shown by, for all $f \in \mathcal{H}$,

$$\begin{aligned}
\|f\|_\infty &= \sup_{x\in\mathcal{X}} |f(x)| \\
&= \sup_{x\in\mathcal{X}} |\langle \kappa(\cdot, x), f\rangle_\mathcal{H}| \\
&\leq \sup_{x\in\mathcal{X}} \|\kappa(\cdot, x)\|_\mathcal{H}\|f\|_\mathcal{H} \\
&= \sup_{x\in\mathcal{X}} \kappa(x, x)\|f\|_\mathcal{H} \\
&\leq \|f\|_\mathcal{H}.
\end{aligned}$$

$\square$

We need one more lemma to connect $\Phi(\hat{f}_\lambda; \hat{P}_n)$ with $\Phi(\hat{f}_\lambda; P)$.

**Lemma D.3.** *For all $\delta \in (0, 1)$, we have, in addition to the statements of Proposition D.2,*

$$\left|\Phi(\hat{f}_\lambda; \hat{P}_n) - \Phi(\hat{f}_\lambda; P)\right| \leq \frac{8G^2}{\lambda n}\sqrt{\frac{\ln(e^2/\delta)}{2}} + 2G\sqrt{\frac{\ln(2/\delta)}{2n}}$$

*with probability* $1 - 2\delta$.

*Proof.* It follows from the uniform law of large number (Theorem H.5) with the high probability range of $\hat{f}_\lambda$ given by (43). Let $\mathcal{G} := \{\varphi_f \ : \ f \in \mathcal{F}, \|f - f_\lambda^*\|_\mathcal{F} \leq d\}$, where $d := \frac{4G}{\lambda}\sqrt{\frac{\ln(e^2/\delta)}{2n}}$. Now, applying Theorem H.5 with $\delta \leftarrow \delta/2$, $\mathcal{F} \leftarrow \pm\mathcal{G}$ and $D \leftarrow 2G$, we get

$$\sup_{\|f-f_\lambda^*\|_\mathcal{F}\leq d} \left|\langle\varphi_f, \ \hat{P}_n - P\rangle\right| \leq 2\mathfrak{R}_n(\mathcal{G}) + 2G\sqrt{\frac{\ln(2/\delta)}{2n}}$$

with probability $1 - \delta$. Since $\|\hat{f}_\lambda - f_\lambda^*\|_\mathcal{F} \leq d$ with probability $1 - \delta$, we further get by union bound

$$\left|\langle\varphi_{\hat{f}_\lambda}, \ \hat{P}_n - P\rangle\right| \leq 2\mathfrak{R}_n(\mathcal{G} \cup -\mathcal{G}) + G\sqrt{\frac{\ln(2/\delta)}{2n}}.$$

with probability $1 - 2\delta$. Since $\Phi(\hat{f}_\lambda; Q) := \langle\varphi_{\hat{f}_\lambda}, \ Q\rangle$ for all $Q \in \mathscr{M}(\mathcal{X}^3)$ the proof is concluded by

$$\begin{aligned}
\mathfrak{R}_n(\mathcal{G}) &= \mathfrak{R}_n(\{\varphi_f \ : \ f \in \mathcal{F}, \|f - f_\lambda^*\|_\mathcal{F} \leq d\}) \\
&\leq \sum_{j=1}^3 \mathfrak{R}_n(\{\varphi_{f,j} \ : \ f \in \mathcal{F}, \|f - f_\lambda^*\|_\mathcal{F} \leq d\}) \\
&\leq Gd\mathfrak{R}_n(\mathcal{B}_\mathcal{F}(0, 1)) \\
&\leq \frac{Gd}{\sqrt{n}}.
\end{aligned}$$

$\square$

Finally, we are ready to prove Theorem 7.3. Observe

$$(1-\gamma)\left|-\frac{1}{1-\gamma}\Phi(\hat{f}_\lambda;\hat{P}_n) - \|\mathcal{R}_\pi(w)\|_{\mathrm{TV}}\right|$$

$$\leq \left\{\Phi(\hat{f}_\lambda;P) - \inf_{f\in\mathcal{F}}\Phi(f;P)\right\} + |\Phi(\hat{f}_\lambda;\hat{P}_n) - \Phi(\hat{f}_\lambda;P)| \qquad (\because (41))$$

Due to Proposition D.2 and Lemma D.3, we have the following inequalities bounding both terms of the RHS with probability $1-2\delta$: The first term is bounded by

$$\Phi(\hat{f}_\lambda;P) - \inf_{f\in\mathcal{F}}\Phi(f;P)$$

$$\leq \mathcal{L}_0(\hat{f}_\lambda;P) - \inf_{f\in\mathcal{F}}\mathcal{L}_0(f;P) \qquad (\because (37) \text{ with } \Psi_0(f)\geq 0, (42))$$

$$\leq \inf_{f\in\mathcal{F}}\mathcal{L}_\lambda(f;P) - \inf_{f\in\mathcal{F}}\mathcal{L}_0(f;P) + \frac{8G^2\ln(e^2/\delta)}{\lambda n} \qquad (\because \text{Proposition D.2})$$

$$\leq s(d) + \frac{\lambda}{2}d^2 + \frac{8G^2\ln(e^2/\delta)}{\lambda n}$$

for all $d > 0$, where $s(d) := \inf_{\|f\|_\mathcal{F}\leq d}\mathcal{L}_0(f;P) - \inf_{f\in\mathcal{F}}\mathcal{L}_0(f;P)$. The second term is bounded by

$$|\Phi(\hat{f}_\lambda;\hat{P}_n) - \Phi(\hat{f}_\lambda;P)| \leq \frac{8G^2}{\lambda n}\sqrt{\frac{\ln(e^2/\delta)}{2}} + 2G\sqrt{\frac{\ln(2/\delta)}{2n}}. \qquad (\because \text{Lemma D.3})$$

Combining these two inequalities, we get

$$(1-\gamma)\left|-\frac{1}{1-\gamma}\Phi(\hat{f}_\lambda;\hat{P}_n) - \|\mathcal{R}_\pi(w)\|_{\mathrm{TV}}\right|$$

$$\leq s(d) + \frac{\lambda d^2}{2} + \frac{8G^2}{\lambda n}\left\{\ln(e^2/\delta) + \sqrt{\frac{\ln(e^2/\delta)}{2}}\right\} + 2G\sqrt{\frac{\ln(2/\delta)}{2n}}$$

with probability $1-2\delta$. Now, since $\lim_{d\to\infty}s(d) = 0$, taking $d = \lambda^{-1/3}$, we have just shown the following proposition, which directly translates into Theorem 7.3.

**Proposition D.4.** *We have*

$$-\frac{1}{1-\gamma}\Phi(\hat{f}_\lambda;\hat{P}_n) \to \|\mathcal{R}_\pi(w)\|_{\mathrm{TV}}$$

*in probability as $\lambda \to 0$ and $n\lambda \to \infty$.*

## E  DUAL REPRESENTATION OF CONVOLUTION NORM (PROOF OF PROPOSITION 7.4)

We first show the following utility lemma.

**Lemma E.1.** *Let $\|f\|_A$ and $\|f\|_B$ be arbitrary norms of $f \in \mathscr{B}(\mathcal{X})$. Also let $\|f\|_{A\vee B} := \|f\|_A \vee \|f\|_B$ be the norm defined by the maximum of these. Then, for all $P \in \mathcal{M}(\mathcal{X})$, we have*

$$\|P\|_{(A\vee B)^*} \leq \inf_{Q\ll P}\left\{\|P-Q\|_{A^*} + \|Q\|_{B^*}\right\},$$

*where we denote the dual norm of $\|\cdot\|_X$ on $\mathscr{B}(\mathcal{X})$ by $\|P\|_{X^*} := \sup_{f\in\mathscr{B}(\mathcal{X}),\|f\|_X\leq 1}\langle f,\ P\rangle$, $P \in \mathcal{M}(\mathcal{X})$.*

*Moreover, if $|\mathrm{supp}(P)|$ is finite and $\|\cdot\|_A$ and $\|\cdot\|_B$ dominate $\|\cdot\|_\infty$, then the equality is attained.*

*Proof.* Observe

$$
\|P\|_{(A \vee B)^*} = \sup_{\substack{f \in \mathscr{B}(\mathcal{X}) \\ \|f\|_A \leq 1 \\ \|f\|_B \leq 1}} \langle f,\, P \rangle
$$

$$
= \sup_{\substack{f,g \in \mathscr{B}(\mathcal{X}) \\ \|f\|_A \leq 1 \\ \|g\|_B \leq 1}} \inf_{Q \ll P} \{ \langle f,\, P \rangle + \langle g - f,\, Q \rangle \} \qquad (\because Q \text{ as a Lagrange multiplier})
$$

$$
= \sup_{\substack{f,g \in \mathscr{B}(\mathcal{X}) \\ \|f\|_A \leq 1 \\ \|g\|_B \leq 1}} \inf_{Q \ll P} \{ \langle f,\, P - Q \rangle + \langle g,\, Q \rangle \}
$$

$$
\leq \inf_{Q \ll P} \left\{ \sup_{\substack{f \in \mathscr{B}(\mathcal{X}) \\ \|f\|_A \leq 1}} \langle f,\, P - Q \rangle + \sup_{\substack{g \in \mathscr{B}(\mathcal{X}) \\ \|g\|_B \leq 1}} \langle g,\, Q \rangle \right\}
$$

$$
= \inf_{Q \ll P} \{ \|P - Q\|_{A^*} + \|Q\|_{B^*} \},
$$

which proves the inequality.

Now suppose $d := \operatorname{supp}(P)$ is finite and $\|\cdot\|_A$ and $\|\cdot\|_B$ dominate $\|\cdot\|_\infty$. Label each element of $\operatorname{supp}(P)$ by $\{x_j\}_{j=1}^d$. Then, following the same equality as above, we get

$$
\|P\|_{(A \vee B)^*} = \sup_{\substack{a \in \mathcal{B}_A \\ b \in \mathcal{B}_B}} \inf_{Q \ll P} \left\{ \sum_{j=1}^d a_j \{ P(x_j) - Q(x_j) \} + \sum_{j=1}^d b_j Q(x_j) \right\},
$$

where

$$
\mathcal{B}_A := \{ (f(x_j))_{j=1}^d \,:\, f \in \mathscr{B}(\mathcal{X}), \|f\|_A \leq 1 \} \subset \mathbb{R}^d,
$$
$$
\mathcal{B}_B := \{ (g(x_j))_{j=1}^d \,:\, g \in \mathscr{B}(\mathcal{X}), \|g\|_B \leq 1 \} \subset \mathbb{R}^d.
$$

The domination of $A$- and $B$-norms over the uniform norm implies there exists $c < \infty$ such that $\mathcal{B}_A, \mathcal{B}_B \subset c\,\mathcal{U}_d$ where $\mathcal{U}_d := \{ z \in \mathbb{R}^d \,:\, \max_{1 \leq j \leq d} |z_j| \leq 1 \}$ denotes the unit hypercube of $\mathbb{R}^d$. Since $c\,\mathcal{U}_d$ is compact and $\mathcal{B}_A$ and $\mathcal{B}_B$ are closed subsets thereof, they are also compact. Thus, Sion's minimax theorem yields

$$
\|P\|_{(A \vee B)^*} = \sup_{\substack{a \in \mathcal{B}_A \\ b \in \mathcal{B}_B}} \inf_{Q \ll P} \left\{ \sum_{j=1}^d a_j \{ P(x_j) - Q(x_j) \} + \sum_{j=1}^d b_j Q(x_j) \right\}
$$

$$
= \inf_{Q \ll P} \left\{ \sup_{a \in \mathcal{B}_A} a_j \{ P(x_j) - Q(x_j) \} + \sup_{b \in \mathcal{B}_B} \sum_{j=1}^d b_j Q(x_j) \right\}
$$

$$
= \inf_{Q \ll P} \{ \|P - Q\|_{A^*} + \|Q\|_{B^*} \}.
$$

This concludes the proof. $\qquad\square$

For $P \in \mathscr{M}(\mathcal{X})$, define

$$
F_u(P) := \sup_{\substack{f \in \mathscr{B}(\mathcal{X}) \\ \|f\|_\infty \leq 1 \\ \|f\|_{\mathcal{H}} \leq u}} \langle f,\, P \rangle.
$$

The following lemma shows that $F_u(P)$ is equal to the $u$-convolution norm. In other words, it gives the dual representation of the $u$-convolution norm in Proposition 7.4.

**Lemma E.2.** *For all $P \in \mathscr{M}(\mathcal{X})$, we have*

$$
\|P\|_{u,\kappa} = F_u(P),
$$

*where $\|P\|_{u,\kappa}$ is given with respect to Definition 7.1.*

*Proof.* Without loss of generality, we assume $\|P\|_{\mathrm{TV}} = 1$. Let $\hat{P}_n$ be the empirical distribution of $P$ given by Definition H.3.

Let $\mathcal{H}$ be the RKHS associated with the kernel $\kappa$. Let $\|f\|_A = \|f\|_{\mathcal{H}}$ and $\|f\|_B = \|f\|_\infty$ in Lemma E.1 and observe that, for all $P \in \mathcal{M}(\mathcal{X})$,

$$F_u(P) = \|P\|_{(A \vee B)^*},$$
$$\|P\|_{u,\kappa} = \inf_{Q \ll P} \{\|P - Q\|_{A^*} + \|Q\|_{B^*}\},$$

since $\|P\|_{\mathcal{H}^*} = \mathrm{MMD}_\kappa(P)$ and $\|P\|_{\mathrm{TV}} = \|P\|_{\infty^*}$. Then, we have

$$0 \leq \|P\|_{u,\kappa} - F_u(P) \leq \mathbb{E}F_u(\hat{P}_n - P)$$

since

$$
\begin{aligned}
F_u(P) &\leq \|P\|_{u,\kappa} &&(\because \text{Lemma E.1})\\
&\leq \mathbb{E}\|\hat{P}_n\|_{u,\kappa} &&(\because \text{Jensen's ineq with Lemma H.3})\\
&= \mathbb{E}F_u(\hat{P}_n) &&(\because \text{Lemma E.1 with } |\operatorname{supp}(\hat{P}_n)| < \infty)\\
&\leq F_u(P) + \mathbb{E}F_u(\hat{P}_n - P). &&(\because \text{triangle inequality of } F_u(\cdot))
\end{aligned}
$$

The proof is concluded remembering that

$$
\begin{aligned}
0 &\leq \mathbb{E}F_u(\hat{P}_n - P)\\
&\leq 2\mathfrak{R}_n(\{f \in \mathscr{B}(\mathcal{X}) : \|f\|_{\mathcal{H}} \leq u, \|f\|_\infty \leq 1\}) &&(\because \text{Theorem H.5})\\
&\leq 2\mathfrak{R}_n(\{f \in \mathscr{B}(\mathcal{X}) : \|f\|_{\mathcal{H}} \leq u\})\\
&\leq 2u\sqrt{\frac{\sup_{x \in \mathcal{X}} \kappa(x,x)}{n}} \to 0 &&(\because \text{Lemma H.6})
\end{aligned}
$$

as $n \to \infty$, where $\mathfrak{R}_n(\mathcal{F})$ is the maximal Rademacher complexity (Definition H.4) $\qquad \square$

## F   PROOF OF COROLLARY 7.5

Let $\bar{w} := w/\|w\|_\infty$ be the normalization of $w$. By Lemma E.2, we have

$$
\begin{aligned}
\|\bar{w} \odot (\hat{\beta} - \beta)\|_{u,\kappa} &= \sup_{\substack{f \in \mathscr{B}(\mathcal{X}) \\ \|f\|_{\mathcal{H}} \leq u \\ \|f\|_\infty \leq 1}} \langle \bar{w}f, \hat{\beta} - \beta \rangle\\
&\leq \sup_{\substack{f \in \mathscr{B}(\mathcal{X}) \\ \|\bar{w}f\|_{\mathcal{H}_w} \leq u \\ \|\bar{w}f\|_\infty \leq 1}} \langle \bar{w}f, \hat{\beta} - \beta \rangle &&(\because \|\bar{w}f\|_{\mathcal{H}_w} = \|f\|_{\mathcal{H}}, \|\bar{w}f\|_\infty \leq \|f\|_\infty)\\
&\leq \sup_{\substack{g \in \mathscr{B}(\mathcal{X}) \\ \|g\|_{\mathcal{H}_w} \leq u \\ \|g\|_\infty \leq 1}} \langle g, \hat{\beta} - \beta \rangle &&(\because \bar{w}\mathscr{B}(\mathcal{X}) \subset \mathscr{B}(\mathcal{X}))\\
&= \|\hat{\beta} - \beta\|_{u,\kappa_w},
\end{aligned}
$$

where $\mathcal{H}_w$ is the RKHS associated with $\kappa_w(x,y) := \bar{w}(x)\bar{w}(y)\kappa(x,y)$. Similarly, we have

$$
\begin{aligned}
&\|\hat{T}_\pi(\bar{w} \odot \hat{\beta}) - T_\pi(\bar{w} \odot \beta)\|_{u,\kappa}\\
&= \sup_{\substack{f \in \mathscr{B}(\mathcal{X}) \\ \|f\|_{\mathcal{H}} \leq u \\ \|f\|_\infty \leq 1}} \langle f, \hat{T}_\pi(\bar{w} \odot \hat{\beta}) - T_\pi(\bar{w} \odot \beta) \rangle\\
&= \sup_{\substack{f \in \mathscr{B}(\mathcal{X}) \\ \|f\|_{\mathcal{H}} \leq u \\ \|f\|_\infty \leq 1}} \langle \bar{w} \otimes f, \hat{\beta}_\pi^{(2)} - \beta_\pi^{(2)} \rangle\\
&\leq \|\hat{\beta}_\pi^{(2)} - \beta_\pi^{(2)}\|_{u,\kappa_w^{(2)}},
\end{aligned}
$$

where $\beta_\pi^{(2)}, \hat{\beta}_\pi^{(2)} \in \mathcal{M}(\mathcal{X}^2)$ are given by $\mathrm{d}\beta_\pi^{(2)}(x, x') := \mathrm{d}\beta(x)\mathrm{d}T_\pi(x'|x)$ and $\mathrm{d}\hat{\beta}_\pi^{(2)}(x, x') := \mathrm{d}\hat{\beta}(x)\mathrm{d}\hat{T}_\pi(x'|x)$, and $\kappa_w^{(2)}((x, x'), (y, y')) := \bar{w}(x)\bar{w}(y)\kappa(x', y')$. Therefore, we have

$$|\|\hat{\mathcal{R}}_\pi(w)\|_{u,\kappa} - \|\mathcal{R}_\pi(w)\|_{u,\kappa}|$$

$$\leq \|\hat{\iota}_\pi - \iota_\pi\|_{u,\kappa} + \frac{\gamma}{1-\gamma}\|\hat{T}_\pi(w \odot \hat{\beta}) - T_\pi(w \odot \beta)\|_{u,\kappa} + \frac{1}{1-\gamma}\|w \odot (\hat{\beta} - \beta)\|_{u,\kappa}$$

$$\leq \|\hat{\iota}_\pi - \iota_\pi\|_{u,\kappa} + \frac{\gamma\|w\|_\infty}{1-\gamma}\|\hat{\beta}_\pi^2 - \beta_\pi^2\|_{u,\kappa_w^{(2)}} + \frac{\|w\|_\infty}{1-\gamma}\|\hat{\beta} - \beta\|_{u,\kappa_w}.$$

Since $\kappa$, $\kappa_w$ and $\kappa_w^{(2)}$ are all bounded, (19) now implies

$$|\|\hat{\mathcal{R}}_\pi(w)\|_{u,\kappa} - \|\mathcal{R}_\pi(w)\|_{u,\kappa}| = O\left(\frac{\|w\|_\infty}{1-\gamma}\sqrt{\frac{u^2 + \ln(1/\delta)}{n}}\right).$$

Combining this with (18) and take the limit with $u \to \infty$ and $n/u^2 \to \infty$, we get the desired result.

## G  PROOF OF THEOREM 7.6

Note that Theorem 7.3 combined with the compactness of $\mathcal{W}$ and the continuity of $w \mapsto \|\mathcal{R}_\pi(w)\|_{\mathrm{TV}}$ implies that $\texttt{EvaluateDBR}(\mathcal{D}, \mathcal{F}, w)$ converges to $\|\mathcal{R}_\pi(w)\|_{\mathrm{TV}}$ uniformly on $w \in \mathcal{W}$. Thus, it suffice to show the following lemma.

**Lemma G.1.** *We have*

$$\min_{u \in \mathcal{U}} \|\mathcal{R}_\pi(\hat{w}_u)\|_{\mathrm{TV}} \to \min_{w \in \mathcal{W}} \|\mathcal{R}_\pi(w)\|_{\mathrm{TV}}.$$

*Proof.* Note also that Corollary 7.5 combined with the compactness of $\mathcal{W}$ and the continuity of $w \mapsto \|\mathcal{R}_\pi(w)\|_{u,\kappa}$ implies

$$\|\hat{\mathcal{R}}_\pi(w)\|_{u,\kappa} \to \|\mathcal{R}_\pi(w)\|_{\mathrm{TV}}$$

uniformly for all $w \in \mathcal{W}$, under suitable asymptotics of $u$ and $n$ as in Corollary 7.5. In other words, for all $c > 0$ and $\delta \in (0, 1)$, there exists $u_0 \geq 1$ and $p_0 > 0$ such that, for all $u \geq u_0$ and $n \geq p_0 u^2$ such that $\sup_{w \in \mathcal{W}} |\|\hat{\mathcal{R}}_\pi(w)\|_{u,\kappa} - \|\mathcal{R}_\pi(w)\|_{\mathrm{TV}}| \leq c$ with probability $\geq 1 - \delta$. Therefore, taking such a pair $(u, n)$ satisfying $u \in \mathcal{U}$ (which exists by the definition of $\mathcal{U}$), we have

$$\min_{w \in \mathcal{W}} \|\mathcal{R}_\pi(w)\|_{\mathrm{TV}} \leq \min_{u' \in \mathcal{U}} \|\mathcal{R}_\pi(\hat{w}_{u'})\|_{\mathrm{TV}} \qquad (\because \text{restriction of domain})$$

$$\leq c + \|\hat{\mathcal{R}}_\pi(\hat{w}_u)\|_{u,\kappa}$$

$$\leq c + \|\hat{\mathcal{R}}_\pi(w_\star)\|_{u,\kappa} \qquad (\because \text{definition of } \hat{w}_u)$$

$$\leq 2c + \|\hat{\mathcal{R}}_\pi(w_\star)\|_{\mathrm{TV}}$$

$$= 2c + \min_{w \in \mathcal{W}} \|\mathcal{R}_\pi(w)\|_{\mathrm{TV}}$$

with probability $\geq 1 - \delta$. Since $c > 0$ can be arbitrary small, we have just proved the desired result. $\qquad \square$

## H  BASIC DEFINITIONS AND RESULTS

This section presents basic results used in the proof of our results for completeness. The main purpose of this section is to show Proposition D.2.

### H.1  SIGNED MEASURES

We first introduce the absolute value and the positive and negative parts of a signed measure. Recall that $\Sigma$ is the Borel algebra of $\mathcal{X}$.

**Definition H.1** (Absolute value and positive and negative parts of signed measure). *For all $P \in \mathscr{M}(\mathcal{X})$, its absolute value is given by $|P| \in \mathscr{M}(\mathcal{X})$ such that*

$$|P|(E) = \sup_{E_+ + E_- = E} \{P(E_+) - P(E_-)\}.$$

*Moreover, its positive and negative parts are given by $P_\pm := (|P| \pm P)/2 \in \mathscr{M}(\mathcal{X})$.*

The following properties are then immediately seen. We omit the proof since it is trivial from the definitions.

**Lemma H.1.** *The following statements are true.*

1. *$P_\pm$ are nonnegative measures.*

2. *$P = P_+ - P_-$ and $|P| = P_+ + P_-$.*

3. *$P, P_\pm \ll |P|$.*

4. *$\||P|\|_{\mathrm{TV}} = \|P_+\|_{\mathrm{TV}} + \|P_-\|_{\mathrm{TV}} = \|P\|_{\mathrm{TV}}$.*

Next, the sign function of a signed measure is defined with the absolute value.

**Definition H.2** (Sign of signed measure). *For all $P \in \mathscr{M}(\mathcal{X})$, its sign is given by $\mathrm{sign}\, P := \frac{\mathrm{d}P}{\mathrm{d}|P|} \in \mathscr{B}(\mathcal{X})$.*

We note that the essential range of the sign function is bounded to $[-1, 1]$.

**Lemma H.2.** *We have $|(\mathrm{sign}\, P)(x)| \leq 1$ for $|P|$-almost every $x \in \mathcal{X}$.*

*Proof.* It follows from

$$\left| \frac{\mathrm{d}P}{\mathrm{d}|P|} \right| = \left| \frac{\mathrm{d}P_+}{\mathrm{d}|P|} - \frac{\mathrm{d}P_-}{\mathrm{d}|P|} \right| \leq \frac{\mathrm{d}P_+}{\mathrm{d}|P|} + \frac{\mathrm{d}P_-}{\mathrm{d}|P|} = \frac{\mathrm{d}|P|}{\mathrm{d}|P|} = 1. \qquad (|P|\text{-almost everywhere})$$

$\square$

Finally, we define the empirical distribution for signed measures.

**Definition H.3** (Empirical distribution of signed measure). *For all $P \in \mathscr{M}(\mathcal{X})$ such that $\|P\|_{\mathrm{TV}} = 1$, we define its $n$-th empirical distribution by*

$$\hat{P}_n := \frac{1}{n} \sum_{i=1}^{n} (\mathrm{sign}\, P)(x_i) \delta_{x_i},$$

*where $\{x_i\}_{i=1}^{n}$ is $n$-i.i.d. sample drawn independently from $|P|$, which is a probability distribution.*

Note that it coincides with the empirical distribution of probability measures if $P$ is nonnegative. One of its most basic properties is the unbiasedness.

**Lemma H.3** (Unbiasedness). *For all $P \in \mathscr{M}(\mathcal{X})$ such that $\|P\|_{\mathrm{TV}} = 1$, we have*

$$P(E) = \mathbb{E}\hat{P}_n(E)$$

*for all $E \in \Sigma$.*

*Proof.* It follows from that, for all $E \in \Sigma$,

$$\begin{aligned}
\mathbb{E}\hat{P}_n(E) &= \frac{1}{n} \sum_{i=1}^{n} \mathbb{E}[(\mathrm{sign}\, P)(x_i)\mathbf{1}\{x_i \in E\}] \\
&= (\mathrm{sign}\, P \odot |P|)(E) && (\because x_i \sim |P|) \\
&= P(E). && (\because \mathrm{sign}\, P = \mathrm{d}P/\mathrm{d}|P|)
\end{aligned}$$

$\square$

## H.2 RADEMACHER COMPLEXITY AND UNIFORM LAW OF LARGE NUMBER

The Rademacher complexity is a measure of the complexity of function class. It is mainly utilized to establish the concentration of the empirical processes corresponding to the function classes, *i.e.*, the uniform law of large number. Throughout the section, let $\boldsymbol{\sigma}^n := \{\sigma_i\}_{i=1}^n$ be a sequence of Rademacher random variables, each of which takes $\pm 1$ with probability $1/2$ independently.

**Definition H.4** (Rademacher complexity). *For a subset of $n$-dimensional vectors $\Theta \subset \mathbb{R}^n$, the Rademacher complexity of $\Theta$ is defined by*

$$\mathfrak{R}(\Theta) := \mathbb{E}_{\boldsymbol{\sigma}^n} \left[ \sup_{\theta \in \Theta} \sum_{i=1}^n \sigma_i \theta_i \right].$$

*Moreover, for $S \in \mathcal{X}^n$ and $\mathcal{F} \subset \mathscr{B}(\mathcal{X})$, the empirical Rademacher complexity of $\mathcal{F}$ with respect to $S$ is defined by*

$$\mathfrak{R}_S(\mathcal{F}) := \mathfrak{R}(\mathcal{F}(S)/n),$$

*where $\mathcal{F}(S) := \{(f(x_1), ..., f(x_n)) \in \mathbb{R} : S = \{x_i\}_{i=1}^n, f \in \mathcal{F}\}$ denotes the set of vectors obtained by applying $f \in \mathcal{F}$ on each element of $S$. Also, we define the $n$-th maximal Rademacher complexity of $\mathcal{F}$ by*

$$\mathfrak{R}_n(\mathcal{F}) := \sup_{S \in \mathcal{X}^n} \mathfrak{R}_S(\mathcal{F}).$$

The following lemma is useful to bound the Rademacher complexity of the composition of functions.

**Lemma H.4** (Rademacher contraction lemma). *For all $\Theta \in \mathbb{R}^n$ and a family of 1-Lipschitz continuous functions $(\varphi_i)_{i=1}^n$, $\varphi_i : \mathbb{R} \to \mathbb{R}$, we have*

$$\mathfrak{R}(\varphi(\Theta)) \le \mathfrak{R}(\Theta),$$

*where $\varphi(\Theta) := \{(\varphi_1(\theta_1), ..., \varphi_n(\theta_n)) \in \mathbb{R}^n : \theta \in \Theta\}$.*

*Proof.* Observe that

$$\mathfrak{R}(\varphi(\Theta)) = \mathbb{E}_{\boldsymbol{\sigma}^n} \sup_{\theta \in \Theta} \sum_{i=1}^n \sigma_i \varphi_i(\theta_i)$$

$$= \frac{1}{2} \mathbb{E}_{\boldsymbol{\sigma}^{n-1}} \left[ \sup_{\theta \in \Theta} \left\{ \sum_{i=1}^{n-1} \sigma_i \varphi_i(\theta_i) + \varphi_n(\theta_n) \right\} + \sup_{\theta' \in \Theta} \left\{ \sum_{i=1}^{n-1} \sigma_i \varphi_i(\theta_i') - \varphi_n(\theta_n') \right\} \right].$$

Since the expression inside the expectation is bounded by

$$\sup_{\theta \in \Theta} \left\{ \sum_{i=1}^{n-1} \sigma_i \varphi_i(\theta_i) + \varphi_n(\theta_n) \right\} + \sup_{\theta' \in \Theta} \left\{ \sum_{i=1}^{n-1} \sigma_i \varphi_i(\theta_i') - \varphi_n(\theta_n') \right\}$$

$$= \sup_{\theta, \theta' \in \Theta} \left\{ \sum_{i=1}^{n-1} \sigma_i \{\varphi_i(\theta_i) + \varphi_i(\theta_i')\} + \{\varphi_n(\theta_n) - \varphi_n(\theta_n')\} \right\}$$

$$\le \sup_{\theta, \theta' \in \Theta} \left\{ \sum_{i=1}^{n-1} \sigma_i \{\varphi_i(\theta_i) + \varphi_i(\theta_i')\} + |\theta_n - \theta_n'| \right\}$$

$$= \sup_{\theta, \theta' \in \Theta} \left\{ \sum_{i=1}^{n-1} \sigma_i \{\varphi_i(\theta_i) + \varphi_i(\theta_i')\} + \theta_n - \theta_n' \right\} \qquad (\because \text{Symmetry of } \theta \text{ and } \theta')$$

$$= \sup_{\theta \in \Theta} \left\{ \sum_{i=1}^{n-1} \sigma_i \varphi_i(\theta_i) + \theta_n \right\} + \sup_{\theta' \in \Theta} \left\{ \sum_{i=1}^{n-1} \sigma_i \varphi_i(\theta_i') - \theta_n' \right\},$$

we have

$$\mathfrak{R}(\varphi(\Theta)) \le \frac{1}{2} \mathbb{E}_{\boldsymbol{\sigma}^{n-1}} \left[ \sup_{\theta \in \Theta} \left\{ \sum_{i=1}^{n-1} \sigma_i \varphi_i(\theta_i) + \theta_n \right\} + \sup_{\theta' \in \Theta} \left\{ \sum_{i=1}^{n-1} \sigma_i \varphi_i(\theta_i') - \theta_n' \right\} \right]$$

$$= \mathfrak{R}(\tilde{\varphi}(\Theta)),$$

where $\tilde{\varphi} = (\varphi_1, ..., \varphi_{n-1}, I)$ and $I$ is the identity map. Iterating the same argument to swap $\varphi_j$ with $I$ for all $j = 1, ..., n - 1$, we get the desired result. $\qquad \square$

The following theorem gives a sufficient condition for the concentration of the empirical process $f \mapsto \langle f, \ \hat{P}_n - P \rangle$, $f \in \mathcal{F}$, with respect to the Rademacher complexity $\mathfrak{R}_n(\mathcal{F})$.

**Theorem H.5** (Uniform law of large number)**.** *For all probability measures $P \in \mathcal{M}(\mathcal{X})$ and all $\mathcal{F} \subset \mathcal{B}(\mathcal{X})$, we have*

$$\mathbb{E} \sup_{f \in \mathcal{F}} \langle f, \ \hat{P}_n - P \rangle \leq 2\mathfrak{R}_n(\mathcal{F}).$$

*Furthermore, we have*

$$\sup_{f \in \mathcal{F}} \langle f, \ \hat{P}_n - P \rangle \leq 2\mathfrak{R}_n(\mathcal{F}) + D\sqrt{\frac{\ln(1/\delta)}{2n}}$$

*with probability $1 - \delta$, where $D := \sup_{f \in \mathcal{F}, x, y \in \mathcal{X}} \{f(x) - f(y)\}$. Here, $\hat{P}_n$ is the empirical distribution of $P$ given by Definition H.3.*

*Proof.* Let $\{x_i\}_{i=1}^n$ and $\{x_i'\}_{i=1}^n$ are two $n$-i.i.d. sample drawn independently from $P$. The first result follows from

$$
\begin{aligned}
\mathbb{E} \sup_{f \in \mathcal{F}} \langle f, \ \hat{P}_n - P \rangle = \mathbb{E} \sup_{f \in \mathcal{F}} \mathbb{E}\left[\frac{1}{n}\sum_{i=1}^n \{f(x_i) - f(x_i')\} \ \middle| \ \{x_i\}_{i=1}^n\right] && (\because \text{Lemma H.3}) \\
\leq \mathbb{E}\left[\sup_{f \in \mathcal{F}} \frac{1}{n}\sum_{i=1}^n \{f(x_i) - f(x_i')\}\right] && \\
= \mathbb{E}\left[\sup_{f \in \mathcal{F}} \frac{1}{n}\sum_{i=1}^n \sigma_i \{f(x_i) - f(x_i')\}\right] && (\because \text{symmetry of } x_i \text{ and } x_i') \\
\leq 2\mathbb{E}\left[\sup_{f \in \mathcal{F}} \frac{1}{n}\sum_{i=1}^n \sigma_i f(x_i)\right] && \\
\leq 2\mathfrak{R}_n(\mathcal{F}). &&
\end{aligned}
$$

To show the second result, define

$$A(S) := \sup_{f \in \mathcal{F}} \left\{\frac{1}{n}\sum_{i=1}^n f(x_i) - \langle f, \ P \rangle\right\}$$

for $S := \{x_i\}_{i=1}^n \in \mathcal{X}^n$ and observe $A(S)$ follows the same law as

$$\sup_{f \in \mathcal{F}} \langle f, \ \hat{P}_n - P \rangle.$$

Thus, it suffices to establish

$$A(S) - \mathbb{E}A(S) \leq D\sqrt{\frac{\ln(1/\delta)}{2n}}$$

with probability $1 - \delta$, which follows from McDiarmid's inequality (Boucheron et al., 2003) applied on $A(S)$. Here, the assumption of McDiarmid's inequality we need to verify is that $A(S') - A(S) \leq D/n$ for all $S' := \{x_i'\}_{i=1}^n \in \mathcal{X}^n$ that only differs from $S$ at the $j$-th element, $1 \leq j \leq n$. This is verified by

$$
\begin{aligned}
A(S') = \sup_{f \in \mathcal{F}} \left\{\frac{1}{n}\sum_{i=1}^n f(x_i') - \langle f, \ P \rangle\right\} \\
= \sup_{f \in \mathcal{F}} \left\{\frac{1}{n}\sum_{i=1}^n f(x_i) - \langle f, \ P \rangle + \frac{1}{n}\{f(x_i') - f(x_i)\}\right\} \\
\leq A(S) + \frac{1}{n}\sup_{f \in \mathcal{F}} \{f(x_i') - f(x_i)\} \\
\leq A(S) + \frac{D}{n}.
\end{aligned}
$$

This concludes the proof. $\square$

### H.3 Reproducing Kernel Hilbert Space

Throughout this section, we assume $\mathcal{H}$ is the reproducing kernel Hilbert space generated with a continuous, symmetric, positive-definite kernel $\kappa : \mathcal{X}^2 \to \mathbb{R}$. Also let $\| \cdot \|_{\mathcal{H}}$ and $\langle \cdot, \ \cdot \rangle_{\mathcal{H}}$ be the associated norm and inner product, and let $\mathcal{B}_{\mathcal{H}}(0, 1) := \{f \in \mathcal{H} \ : \ \|f\|_{\mathcal{H}} \le 1\}$ be the closed unit ball of $\mathcal{H}$. The following lemma shows the Rademacher complexity of RKHS can be explicitly bounded. Note that $c_0$-universal kernel is uniformly bounded and hence $\|\kappa\|_\infty < \infty$.

**Lemma H.6** (Rademacher complexity of RKHS). *We have*

$$\mathfrak{R}_n(\mathcal{B}_{\mathcal{H}}(0,1)) \le \sqrt{\frac{\|\kappa\|_\infty}{n}}.$$

*Proof.* It follows from that, for all $S \in \mathcal{X}^n$,

$$
\begin{aligned}
\mathfrak{R}_S(\mathcal{B}_{\mathcal{H}}(0,1)) &= \mathbb{E}_{\boldsymbol{\sigma}^n}\left[ \sup_{f \in \mathcal{B}_{\mathcal{H}}(0,1)} \frac{1}{n} \sum_{i=1}^{n} \sigma_i f(x_i) \right] \\
&= \mathbb{E}_{\boldsymbol{\sigma}^n}\left[ \sup_{f \in \mathcal{B}_{\mathcal{H}}(0,1)} \frac{1}{n} \sum_{i=1}^{n} \sigma_i \langle \kappa(\cdot, x_i), \ f \rangle_{\mathcal{H}} \right] \\
&= \mathbb{E}_{\boldsymbol{\sigma}^n}\left\| \frac{1}{n} \sum_{i=1}^{n} \sigma_i \kappa(\cdot, x_i) \right\|_{\mathcal{H}} \\
&\le \sqrt{ \mathbb{E}_{\boldsymbol{\sigma}^n}\left\| \frac{1}{n} \sum_{i=1}^{n} \sigma_i \kappa(\cdot, x_i) \right\|_{\mathcal{H}}^2 } \qquad (\because \text{Jensen's ineq.}) \\
&= \sqrt{ \mathbb{E}_{\boldsymbol{\sigma}^n} \left\langle \frac{1}{n} \sum_{i=1}^{n} \sigma_i \kappa(\cdot, x_i), \ \frac{1}{n} \sum_{j=1}^{n} \sigma_j \kappa(\cdot, x_j) \right\rangle_{\mathcal{H}} } \\
&= \sqrt{ \frac{1}{n^2} \sum_{i=1}^{n} \langle \kappa(\cdot, x_i), \ \kappa(\cdot, x_i) \rangle_{\mathcal{H}} } \\
&= \sqrt{ \frac{1}{n^2} \sum_{i=1}^{n} \kappa(x_i, x_i) } \\
&\le \sqrt{ \frac{\sup_{x \in \mathcal{X}} \kappa(x, x)}{n} }.
\end{aligned}
$$

$\square$

The following lemma shows the compactness of the closed RKHS balls in the $L^2$-metrics.

**Lemma H.7** (Compactness of RKHS). *$\mathcal{B}_{\mathcal{H}}(0, 1)$ is compact with respect to $L^2(P)$ for all positive measures $P \in \mathscr{M}(\mathcal{X})$.*

*Proof.* Mercer's theorem gives an eigen decomposition of $\kappa$ such that

$$\kappa(x, y) = \sum_{k=1}^{\infty} \lambda_k \phi_k(x) \phi_k(y),$$

where the convergence is uniform on $\mathcal{X}^2$, $\{\phi_k : \mathcal{X} \to \mathbb{R}\}_{k=1}^{\infty}$ is a continuous orthonormal basis of $L^2(P)$ and $\Lambda := \{\lambda_k \in \mathbb{R}_{\ge 0}\}_{k=1}^{\infty}$ is a nonnegative decreasing sequence with $\lim_{k \to \infty} \lambda_k = 0$. It then follows from the standard function analysis that $\mathcal{B}_{\mathcal{H},1}$ under the $L^2(P)$-metric is isometric to $U(\Lambda) := \{a \equiv (a_k)_{k=1}^{\infty} \ : \ \sum_{k \ge 0: a_k \ne 0} a_k^2 / \lambda_k \le 1\}$ under the $\ell^2$-metric via the mapping $a_k(f) := \int f(x)\phi_k(x)\mathrm{d}P(x)$. Here, we evaluate $0/0 = \infty$. Therefore, the compactness of $\mathcal{B}_{\mathcal{H},1}$ in $L^2(P)$ follows from the compactness of $U(\Lambda)$ in $\ell^2$. To show the compactness of the latter, it

suffices to show its completeness and total boundedness. The completeness is trivial, while the total boundedness follows from the fact that the elements of $U(\Lambda)$ is approximated with their projections onto the first $K$ coordinates where the approximation error is bounded by $\lambda_{K+1}$, which tends to zero as $K \to \infty$. $\qquad\square$

