# OpenReview forum: "Interval-based Offline Policy Evaluation without Sufficient Exploration or Realizability"
_ICLR.cc/2023/Conference — Submitted to ICLR 2023_

### Official Review · Reviewer_Zijs · 2022-10-17

**Confidence:** 3
**Clarity, Quality, Novelty And Reproducibility:** The paper is well-written. The proof …
**Correctness:** 4
**Technical Novelty And Significance:** 3
**Empirical Novelty And Significance:** Not applicable
**Recommendation:** 6

**Strength And Weaknesses:**

### Strength

The method and analysis in this paper is interesting and inspiring. The new problem setup minimax-bias OPI is well-motivated, and it's nice to see their methods does not require the common assumptions in previous literatures to ensure the validity.


### Weakness

The final results in Sec. 7 is just an asymptotic guarantee and it is unclear about the convergence speed. Especially, it's unclear how well $||P||\_u$ approximates $||P||\_{TV}$ when $u$ is large but not infinite.

**Summary Of The Paper:**

This paper studied policy value interval estimation problem given offline data without sufficient coverage and without realizability assumptions. The authors showed that without coverage assumption there would be an ``irreducible bias'' and therefore, proposed a new problem setup called minimax-bias OPI, which targets at estimating (near-)optimal (shortest) interval covering the true value $J(\pi)$ with a shift at most at the level of the minimax bias. After that, in Sec. 6, they suggested an objective function to compute a near-optimal interval, which is also robust to the misspecification error of function class. Finally, in Sec. 7, they proposed how to computably approximate the solution of the objective function and established asymptotic guarantees.

**Summary Of The Review:**

This paper has many interesting results. Although there is still some limitation, I would suggest the acceptance of this paper.

---

> ### Author Response · Authors · 2022-11-14
> **Thank you**
>
> Thank you for your encouraging feedbacks!
> We agree that the asymptoticity of the current results is one of the limitations and would like to state it more clearly as a concluding remark.

---

> > ### Comment · Reviewer_Zijs · 2022-12-08
> > **Thanks for the rebuttal**
> >
> > Thanks for the response. After reading the review by other reviewers, I also think the asymptotic guarantee (like Thm. 7.6) is kind of weak and it would be better to have some empirical verification to show their method can indeed work. Given that I still find some of the parts in this paper is somewhat new and interesting, I plan to slightly lower down my score to 6.

---

### Official Review · Reviewer_HEuP · 2022-10-20

**Confidence:** 3
**Correctness:** 4
**Technical Novelty And Significance:** 2
**Empirical Novelty And Significance:** Not applicable
**Recommendation:** 3

**Clarity, Quality, Novelty And Reproducibility:**

Clarity: This paper is clear in general.Theorems have no ambiguity to me. However, I do not see enough explanation of the results. This is a mathematically heavy paper, and therefore it is necessary to give explanations for its results and definitions.

Quality: The mathematics seems rigorous, though I did not check all the details.

Novelty: The technical novelty is unclear given the current presentation. The paper does not compare its bound with related works either theoretically or empirically.

Reproducibility: There is no empirical results to reproduce.


**Strength And Weaknesses:**

Strength:
The problem itself is very meaningful and interesting. The setting where the commonly assumed coverage and realizability fail is very practical and worth investigation.
The theoretical results are well-structured. The role of each section looks clear.

---
Weakness:
1. This paper fails to discuss a series of works that are very related to this one. OPE is an extensively considered problem studied by a variety of authors. Almost every part of this paper has some very related works. To be specific,
(i) It is already a well-known result that OPE incurs bias without the coverage assumption (e.g. [1]). [2] has conducted a bias as well as variance analysis for OPE of tabular MDPs years ago. The relation between the lower bound in this paper and hardness results from these papers should be discussed as they seem closely related to me.
(ii) IS/MIS is a common technique for OPE, studied by, [3-5], to list a few. In particular, [3] gives the sharpest bound for tabular MDP under the coverage assumption. How does the bound compare to this paper?
(iii) Interval estimation is great. However, for some papers with point estimation, the results can be transformed into high-probability intervals (see for e.g. [7, 8] which considered the linear function case). [10] also consider OPE like this paper. Although this paper focus on the setting without coverage assumption, it is still necessary to investigate that, If under the assumption of coverage and realizability, how do these results compare to the interval in this paper?

2. There is no numerical simulation to justify the theory. It seems necessary to show exactly how this paper compares to other IS-based OPE methods for tabular setting with coverage assumption (e.g. [3]).
Mathematics is heavy, but there is not enough explanation of the implication behind the theory. For example, why do we need two variants of Bellman equation; in the tabular setting, if the final bound does not depend on the cardinality of the state-action space, then which term characterizes the complexity.


---

[1] Wang, Ruosong, et al. "Instabilities of offline rl with pre-trained neural representation." International Conference on Machine Learning. PMLR, 2021.

[2] Mannor, Shie, et al. "Bias and variance in value function estimation." Proceedings of the twenty-first international conference on Machine learning. 2004.

[3] Yin, Ming, and Yu-Xiang Wang. "Asymptotically efficient off-policy evaluation for tabular reinforcement learning." International Conference on Artificial Intelligence and Statistics. PMLR, 2020.

[4] Li, Lihong, et al. "Unbiased offline evaluation of contextual-bandit-based news article recommendation algorithms." Proceedings of the fourth ACM international conference on Web search and data mining. 2011.

[5] Li, Lihong, Rémi Munos, and Csaba Szepesvári. "Toward minimax off-policy value estimation." Artificial Intelligence and Statistics. PMLR, 2015.

[6] Thomas, Philip, and Emma Brunskill. "Data-efficient off-policy policy evaluation for reinforcement learning." International Conference on Machine Learning. PMLR, 2016.

[7] Duan, Yaqi, Zeyu Jia, and Mengdi Wang. "Minimax-optimal off-policy evaluation with linear function approximation." International Conference on Machine Learning. PMLR, 2020.

[8] Min, Yifei, et al. "Variance-aware off-policy evaluation with linear function approximation." Advances in neural information processing systems 34 (2021): 7598-7610.

[10] Hao, Botao, et al. "Bootstrapping Fitted Q-Evaluation for Off-Policy Inference." International Conference on Machine Learning. PMLR, 2021.

**Summary Of The Paper:**

This paper studies offline policy interval evaluation for discounted MDPs where the goal is to generate an interval that contains the ground truth with high probability. Specifically, this paper shows that, without the coverage and realizability, there exists a lower bound of the asymptotic bias. Then the paper proposes an interval estimation method that can output an asymptotically valid interval without the coverage and realizability assumptions. When the realizability assumption holds, the proposed interval is shown to be optimal.

**Summary Of The Review:**

My current recommendation is reject.
The paper indeed considers an important problem and a meaningful setting. But my major concern is that the contribution and significance of the paper is unclear.

The details are as follows:
There is a lack of comparison with closely related work.
Not enough explanation of the results. It makes this paper hard to understand for readers who are not very familiar with this exact topic.
No empirical evidence to show the validity of the proposed algorithm.
Overall, I think a significant amount of modification is needed.

---

> ### Author Response · Authors · 2022-11-14
> **Thank you**
>
> Thank you for your helpful feedbacks!
> In particular, we are grateful for the extensive list of OPE references you have kindly provided.
>
> > This paper fails to discuss a series of works that are very related to this one.
>
> We consider we have covered the necessary portion of related work to clarify our contributions to the literatures of OPE (Section 1) and OPI (Section 2).
> However, as pointed out by the reviewer, we could better highlight our contributions to the literature of OPE, and we would like to discuss it in this response.
>
> The main contribution to the literature of OPE, which the reviewer is mainly concerned about, is that we provide an alternative formulation to evaluate the value of a given policy based on **interval estimation** (OPI) when accurate **point estimation** (OPE) is not possible, i.e. when neither of the sufficient exploration nor the realizability hold.
> In other words, our objective is to cope with the difficult case where the ordinary OPE methods fail, and our means to achive the objective is OPI.
> Therefore, we believe it is reasonable to focus on OPI methods to discuss related work.
> Note that we do not argue that our method performs better than existing OPE methods when both assumptions hold; in such cases, we believe existing OPE methods would perform well, and there is no reason to consider the minimax bias, which becomes zero.
>
> Below, let us discuss more closely how the previous studies you mentioned are related to ours:
>
> - [1] can be considered as more empirical-oriented study on the problem of the insufficient exploration and the unrealizability. In fact, the same (first and last) authors published [11] in ICLR of the same year, which is more theory-oriented and relevant to ours (we cited it).
> - Other than the implicit assumptions of the sufficient exploration and the realizability, the major difference between [2] and ours is that they study the bias/variance of specific estimators, whereas we study theoretical lower bound independent of estimators.
> - As for the comparison with the existing IS/MIS-related studies [3-6], the idea of estimating the projected importance weight $w^\sharp\_\pi$ is completely new. This difference is essential in the sense it makes us free from the assumption of sufficient exploration unlike the previous IS/MIS approaches.
> - Both of [7,8] require quite strong conditions on the exploration and the realizability. Thus, their aims are of independent interest to ours as discussed above.
> - We actually cited [10], featuring its exactness of the interval estimation. See Section 2 for the details of the comparison.
>
> > For example, why do we need two variants of Bellman equation
>
> We only need the distributional one.
> The functional/ordinary Bellman equation is put there just for comparison.
> We will make it clearer or remove the functional one for simplicity.
>
> > in the tabular setting, if the final bound does not depend on the cardinality of the state-action space, then which term characterizes the complexity.
>
> The complexity of statistical estimation is incorporated implicitly as the convergence rate of the asymptotic limits.
> Its explicit evaluation is left open for future work.
> This is because our primary focus is on formalizing the theoretical limit in the assumption-less setting and
> investigate if it is possible at all to estimate that theoretical limit in the first place.
>
>
> ### Reference:
>
> [1-8,10] *Borrowed from the original review*
> [11] Wang, R., Foster, D. P., and Kakade, S. M. (2020). What are the statistical limits of offline rl with linear function approximation? arXiv preprint arXiv:2010.11895.

---

### Official Review · Reviewer_QpFW · 2022-10-23

**Confidence:** 4
**Correctness:** 4
**Technical Novelty And Significance:** 2
**Empirical Novelty And Significance:** Not applicable
**Recommendation:** 5

**Clarity, Quality, Novelty And Reproducibility:**

As I mentioned above, it would be great to present the minimax bias in a more in-depth manner. In addition, I feel that the assumptions about function class should be presented explicitly. Other than that, I feel that the presentation is pretty clear.

The novelty seems not about theoretical analysis, but about proposing a new notion that captures the effect of insufficient data coverage in OPE. I feel that this notion can be quite impactful, but the current theory and understanding seems not deep enough.

**Strength And Weaknesses:**

Strength: This paper studies a theoretical problem that is more realistic in the sense that it does not impose a coverage assumption over the dataset. The notion of minimax bias seems interesting.

Weaknesses: I feel that this paper can be improved in the following aspects.

1. It would be great to have a more in-depth exposition of the minimax bias. First, it would be better to have a rigorous definition in the “definition” environment. Second, explain why it appears. Third, give a concrete example (maybe on a toy problem like bandit) to explain it.

2. The authors seem to assume that $w^\sharp$ lies in a certain class such that the last term in (8) vanishes. The authors should state this matter clearly.

3. Although the results seem interesting, I am afraid that the bound might not be useful. In particular, the TV-norm of $R_{\pi}$ seems a very conservative upper bound because $1/ (1- \gamma)$ is a huge number. That means, if $ R_{\pi}$ is not small, the confidence region can be vacuous.

4. The theoretical analysis seems not so well developed. First, the TV-norm is not the exact thing used in estimation – an approximation via convolution norm is employed. We only know such an approximation is accurate when $u$ is sufficiently large. But how large should we need? What is the error when $u$ and $n$ are both finite? Second, the error analysis is merely asymptotic. From standard concentration results, one should be able to develop a non-asymptotic theory.




**Summary Of The Paper:**

This paper studies the problem of offline policy evaluation (OPE) in the reinforcement learning setting without assumption that the offline data is well-explored. For such a problem, this paper characterizes the minimax estimation bias caused by the data distribution having a limited support. Moreover, the TV-norm of the Bellman error of the visitation measure gives an upper and lower bound of the bias. Based on that, this paper proposes an estimation method for estimating the bias, which is proved to be consistent.

**Summary Of The Review:**

This paper proposes a notion that characterizes the insufficient coverage of offline data in OPE. It also proposes a method to estimate such a quantity, which leads to a procedure to estimate a confidence region of the expected rewards.

---

> ### Author Response · Authors · 2022-11-14
> **Thank you for your helpful feedbacks and suggestions**
>
> > 1. It would be great to have a more in-depth exposition of the minimax bias. First, it would be better to have a rigorous definition in the “definition” environment. Second, explain why it appears. Third, give a concrete example (maybe on a toy problem like bandit) to explain it.
>
> We agree that the minimax bias deserves its own definition environment and will do so.
> On the 'why' explanation and the concrete example, we believe that the construction of the worst-case environments $\mathcal{M}\_{\pm}$ in the proof (sketch) of Theorem 4.1 can serve as an illustrative example.
> In short, the minimax bias occurs since one has no access to the state-transition dynamics nor the reward function outside of the data support.
> We would like to include such intuitive explanation in the main text.
>
> > 2. The authors seem to assume that $w\_\pi^\sharp$ lies in a certain class such that the last term in (8) vanishes. The authors should state this matter clearly.
>
> In fact, we do not explicitly assume the realizability $w\_\pi^\sharp\in \mathcal{W}$.
> We instead present all of our result taking into account the realizability error $\epsilon\_{\mathcal{W}}$, defined just after (12).
> (Also note that $w_\star$ is the **in-model** optimal weight function, which may be different from $w\_\pi^\sharp$).
>
> > 3. Although the results seem interesting, I am afraid that the bound might not be useful. In particular, the TV-norm of $\mathcal{R}\_\pi$ seems a very conservative upper bound because $1/(1-\gamma)$ is a huge number. That means, if $\mathcal{R}\_\pi$ is not small, the confidence region can be vacuous.
>
> We have two arguments respectively for the vacuousness and the usefulness.
>
> First, since the TV norm $\\|\mathcal{R}\_\pi(w)\\|\_{TV}$ becomes 1 with $w=0$ regardless of the value of $\gamma$, the resulting bound is never vacuous in the strict sense as long as the function approximator $\mathcal{W}$ contains the zero function as in the case of linear/kernel function classes or the neural-network-based ones.
> Moreover, when you employ a function approximator expressive enough to make $\epsilon\_{\mathcal{W}}/(1-\gamma)$ small,
> the bound matches the minimax bias as shown by (12), which is tight by definition in the worst-case scenario.
> In practice, the function approximator would lie somewhere in between the two extrema: the one only containing the zero function and the one makes $\epsilon\_{\mathcal{W}}/(1-\gamma)$ negligible.
> Therefore, our TV-norm-based bound is unlikely to be vacuous.
>
> Second,
> even if it ends up being too large to be useful as a policy-value estimate,
> we argue that the bound is still useful in another sense.
> Here is why:
> Since the minimax bias is **unavoidable** in the worst case without additional source of information (i.e., domain knowledge),
> the prohibitively large size of the minimax-bias estimate can be considered as a useful piece of information.
> In particular, it suggests that the users should either collect more data to reduce the minimax bias or
> stop using domain-knowledge-free approaches (like employing neural networks as universal learners) and seek for reliable source of domain knowledge.
>
>
> > 4. The theoretical analysis seems not so well developed. First, the TV-norm is not the exact thing used in estimation – an approximation via convolution norm is employed. We only know such an approximation is accurate when $u$ is sufficiently large. But how large should we need? What is the error when $u$ and $n$ are both finite? Second, the error analysis is merely asymptotic. From standard concentration results, one should be able to develop a non-asymptotic theory.
>
> Thank you for pointing this out. This is one of the major limitations of this work and we will state it clearer.
> That being said, the standard results of the nonasymptotic concentration theory usually require the boundedness of the statistical complexity of the class of the estimation target -- in this case, the weight function $w$ (or the associated function $f$ that attains the supremum). So, the usual nonasyptotic approaches are inapplicable without the realizability assumption, avoiding which is the motivation of this study to start with.
> A possible middle-ground strategy is to pursuit the adaptivity, i.e., the arguments like "non-asymptotic guarantee X holds if the realizability condition A is satisfied, and asymptotic guarantee Y holds no matter what".
> We would like to leave it open for future study.

---

### Official Review · Reviewer_GGmU · 2022-10-31

**Confidence:** 2
**Correctness:** 3
**Technical Novelty And Significance:** 3
**Empirical Novelty And Significance:** Not applicable
**Recommendation:** 6

**Clarity, Quality, Novelty And Reproducibility:**

As mentioned above, I think the clarity of the paper needs to be improved. The paper has some novelty. Since it is a theoretical paper, I don't see any reproducibility issues.

**Strength And Weaknesses:**

Strength:

1) The problem of estimating the confidence interval of the value of a policy using offline data is very important. Since it is known that when the offline data is not exploratory enough, it is hard to get accurate point estimation, it's important to get good interval estimation.
2) The theoretical analysis of this paper seems to be sound, although I did not check the details.

Weakness:
1) I think the notation in this paper is a bit heavy and I found it a bit hard to understand all the details. I also think it would be good to move the actual algorithms to the main sections of the paper instead of putting them in the appendix.
2) It would be good to have some empirical evaluation.

There are a few places that I don't fully understand:
Page 8 Proof of Proposition 7.4: why does the Rademacher complexity of the unit ball in H of the order O(n^{-1/2})? I assume it should also depend on certain intrinsic dimension of the space H.


**Summary Of The Paper:**

This paper proposes the problem of minimax bias estimation of the value of a policy using offline data. The algorithm is based on marginal-importance-sampling and minimizing the Bellman residual error.

**Summary Of The Review:**

I think this is a good paper overall. But I am not very confident in my review since I did not check all the details.

---

> ### Author Response · Authors · 2022-11-14
> **Thank you**
>
> Thank you for your helpful feedbacks and suggestions!
> We would like to revise the manuscript incorporating your feedbacks on the readability issues with the notation and algorithms.
>
> > Page 8 Proof of Proposition 7.4: why does the Rademacher complexity of the unit ball in H of the order O(n^{-1/2})? I assume it should also depend on certain intrinsic dimension of the space H.
>
> This is because the Rademacher complexity of the unit RKHS ball is bounded by $\sqrt{\frac{\\|\kappa\\|\_\infty}{n}}$
> and we assume the kernel is uniformly bounded, $\\|\kappa\\|\_\infty<\infty$, like the Gaussian kernel. We have made it explicit in the main text
> and added the formal statement as Lemma H.6 in the updated manuscript.

---

> > ### Comment · Reviewer_GGmU · 2022-11-20
> > **Thanks for the clarification**
> >
> > Thanks for the clarification. I'll keep my score although I am not very confident about my evaluation.

---

### Decision · Program_Chairs · 2023-01-20

**Decision:**

Reject

**Justification For Why Not Higher Score:**

Lack of empirical evidence and insufficient theoretical findings.

**Justification For Why Not Lower Score:**

N/A

**Metareview: Summary, Strengths And Weaknesses:**

While the reviewers like the problem studied in the paper and believe there are new and interesting ideas in this work, they do not see the contributions of the paper significant enough for acceptance. The paper is theoretical and there is no experiment to support the findings and to provide a comparison with the existing results. However, the theoretical results are rather limited and do not properly address any of the critical problems in OPE. I agree with the reviewers that the paper is not ready for publication at its current form, but I think it is promising. I would strongly recommend that the authors take the reviewers' comments into account, improve their work either with better theory or a proper mix of theory and empirical evaluation, and prepare it for an upcoming venue.